# Climate predicts geographic and temporal variation in mosquito-borne disease dynamics on two continents

Jamie M. Caldwell [1] ✉, A. Desiree LaBeaud [2], Eric F. Lambin[3,4], Anna M. Stewart-Ibarra [5,6], Bryson A. Ndenga [7], Francis M. Mutuku [8], Amy R. Krystosik [2], Efraín Beltrán Ayala[9], Assaf Anyamba[10], Mercy J. Borbor-Cordova [11], Richard Damoah [12], Elysse N. Grossi-Soyster[2], Froilán Heras Heras[13], Harun N. Ngugi[14,15], Sadie J. Ryan [16,17,18], Melisa M. Shah[19], Rachel Sippy [13,20,21] & Erin A. Mordecai [1]

Climate drives population dynamics through multiple mechanisms, which can lead to seemingly context-dependent effects of climate on natural populations. For climate-sensitive diseases, such as dengue, chikungunya, and Zika, climate appears to have opposing effects in different contexts. Here we show that a model, parameterized with laboratory measured climate-driven mosquito physiology, captures three key epidemic characteristics across ecologically and culturally distinct settings in Ecuador and Kenya: the number, timing, and duration of outbreaks. The model generates a range of disease dynamics consistent with observed *Aedes aegypti* abundances and laboratory-confirmed arboviral incidence with variable accuracy (28–85% for vectors, 44–88% for incidence). The model predicted vector dynamics better in sites with a smaller proportion of young children in the population, lower mean temperature, and homes with piped water and made of cement. Models with limited calibration that robustly capture climate-virus relationships can help guide intervention efforts and climate change disease projections.

[1] Department of Biology, Stanford University, Stanford, CA, USA. [2] Department of Pediatrics, Division of Infectious Diseases, Stanford University, Stanford, CA, USA. [3] School of Earth, Energy & Environmental Sciences, and Woods Institute for the Environment, Stanford University, Stanford, CA, USA. [4] Georges Lemaître Earth and Climate Research Centre, Earth and Life Institute, Université Catholique de Louvain, Louvain-la-Neuve, Belgium. [5] Department of Medicine and Department of Public Health and Preventative Medicine, SUNY Upstate Medical University, Syracuse, NY, USA. [6] InterAmerican Institute for Global Change Research (IAI), Montevideo, Uruguay. [7] Centre for Global Health Research, Kenya Medical Research Institute, Kisumu, Kenya. [8] Department of Environment and Health Sciences, Technical university of Mombasa, Mombasa, Kenya. [9] Technical University of Machala, Machala, Ecuador. [10] Universities Space Research Association and NASA Goddard Space Flight Center, Greenbelt, MD, USA. [11] Facultad de Ingeniería Marítima y Ciencias del Mar, Escuela Superior Politécnica del Litoral, ESPOL, Guayaquil, Ecuador. [12] Morgan State University and NASA Goddard Space Flight Center, Greenbelt, MD, USA. [13] Center for Research SUNY-Upstate-Teófilo Dávila Hospital, Machala, Ecuador. [14] Department of Biological Sciences, Chuka University, Chuka, Kenya. [15] Department of Zoology, School of Biological Sciences University of Nairobi, Nairobi, Kenya. [16] Emerging Pathogens Institute, University of Florida, Gainesville, FL, USA. [17] Quantitative Disease Ecology and Conservation (QDEC) Lab, Department of Geography, University of Florida, Gainesville, FL, USA. [18] School of Life Sciences, University of KwaZulu, Natal, South Africa. [19] Department of Medicine, Division of Infectious Diseases, Stanford University, Stanford, CA, USA. [20] Institute for Global Health and Translational Science, SUNY-Upstate Medical University, Syracuse, NY, USA. [21] Department of Medical Geography, University of Florida, Gainesville, FL, USA. ✉email: jamie.sziklay@gmail.com

Climate is a major driver of species interactions and population dynamics, but the mechanisms underlying the ecological effects of climate are often poorly understood and rarely tested in the field[1]. One of the primary ways that climate impacts populations are through its effects on species' vital rates[2]. However, the effects of climate on population dynamics may appear context-dependent in the field because multiple climate variables can act synergistically, with each climate variable potentially affecting multiple vital rates, and their impacts may be nonlinear, changing direction, and relative importance across a gradient of conditions[3,4]. Therefore, paradoxically, while climate is thought to be one of the most pervasive drivers of ecological processes, its directional and dynamical effects on systems are often poorly understood and difficult to predict. Vector-borne diseases provide an interesting case study to test whether climate-sensitive traits measured in controlled, laboratory settings can reproduce the wide range of dynamics observed in the field. For example, the transmission of mosquito-borne viral (arboviral) diseases such as dengue, chikungunya, and Zika occur along a spectrum from low levels of year-round endemic transmission[5] to large seasonal or interannual outbreaks[6]. We hypothesize that important features of these differing dynamics arise due to regional or seasonal differences in climate, where the magnitude and direction of the effects of climate on vector and disease dynamics differ[7–12].

Understanding the mechanisms that drive disease dynamics can help address two critically important research priorities for arboviruses like dengue, chikungunya, and Zika: assessing intervention strategies and projecting climate change impacts on disease dynamics. While phenomenological models often replicate arboviral disease dynamics remarkably well[13], mechanistic models that do not rely on local data for calibration and capture mosquito population dynamics and interactions between mosquitoes and humans will provide more realistic projections for epidemic dynamics across a broad range of transmission settings. With no widely available vaccine, vector control (e.g., larvicides, Wolbachia-infected mosquito releases) remains the primary method for preventing arboviral disease transmission, and, like other vector-borne diseases with complex transmission dynamics, model simulations can help guide effective intervention efforts[14,15]. Further, mechanistic models are better suited to predict how climate change will impact future disease burden and distribution, as projected climate conditions are outside the current arboviral climate niche space[16]. Despite the potential usefulness of mechanistic approaches, validation with vector and disease data are limited, raising an important question about which epidemic characteristics, if any, we should expect a model to capture when the model was parameterized with data that is on different scales (e.g., individuals versus populations) and independent from the transmission system we wish to predict. Thus, because we cannot study epidemic dynamics in every possible transmission setting, it becomes important to understand the extent to which models derived from fundamental and laboratory-measured traits explain disease dynamics across diverse settings.

We hypothesize that a climate-driven mechanistic model with limited calibration should capture many important characteristics of disease dynamics for dengue, chikungunya, and Zika because of the ecology of *Aedes aegypti*, the primary disease vector. *Ae. aegypti* are anthropophilic, globally distributed mosquitoes that breed in artificial containers with standing water[17,18]. All mosquito and parasite traits that are important for transmission and linked to metabolism, such as reproduction, development, survival, biting rate, and extrinsic incubation period, are temperature-dependent with an intermediate thermal optimum[19–21]. Humidity is positively associated with mosquito

survival because the high surface area to volume ratio of mosquitoes exposes them to desiccation[22,23]. Standing water from rainfall provides essential larval and pupal habitats for mosquitoes, but the relationship is complex because heavy rainfall can flush away breeding habitats[24–26], and water-storage practices during a drought can increase water availability, mosquito abundance, and contact between mosquitoes and people[27–29]. A previous simulation study predicted that in settings with suitable climate for transmission throughout the year (e.g., mean temperature = 25 °C; range = 20–30 °C), temperature drives the timing and duration of outbreaks, but not the maximum number of infections or final epidemic size[30]. This finding suggests that a model that incorporates temperature-dependent vector traits should capture some important epidemic characteristics.

Arboviral dynamics differ considerably in South America and sub-Saharan Africa, potentially because of differences in climate and socio-ecological conditions. Previous studies have found that *Ae. aegypti* and dengue were positively associated with warm and wet conditions in South America and sub-Saharan Africa[6,31–33], although other *Ae. aegypti*-vectored arboviruses in Africa such as chikungunya have been associated with warm and dry conditions[34]. Countries on both continents have all four dengue serotypes circulating and have recently experienced outbreaks of chikungunya; yet, arboviral transmission dynamics differ in each region. In South America, dengue is a re-emerging disease with large seasonal epidemics that frequently result in severe dengue[6]; by contrast, in sub-Saharan Africa, dengue is transmitted at low levels year-round[5] and intermittent self-limiting outbreaks often go undetected[35]. Further, compared with South America, severe dengue is rare in sub-Saharan Africa, perhaps because African strains of *Ae. aegypti* have lower susceptibility to all four dengue serotypes[36], and/or because people of African ancestry are less susceptible to severe dengue[37].

In this work, we test the extent to which climate-driven mosquito traits drive disease dynamics across two geographically distinct regions and characterize additional climatological, ecological, and social factors that may mediate the effects of climate on disease dynamics. We build on previous mechanistic and semi-mechanistic models that incorporate the *Aedes* mosquito life cycle and human disease dynamics[30,38–42] by combining a suite of temperature, humidity, and rainfall-dependent trait functions into one epidemiological model. We validate the model with *Ae. aegypti* abundances and laboratory-confirmed dengue, chikungunya, and Zika cases from two equatorial countries with distinct socioeconomic, geographic, cultural, and disease transmission settings: Ecuador and Kenya. We find that a climate-driven model with limited calibration to local data capture three key epidemic characteristics across diverse settings: the number, timing, and duration of outbreaks. The model generates a range of vector and disease dynamics with varying levels of accuracy. Further, we find that the model predicted vector dynamics better in sites with a smaller proportion of young children in the population, lower mean temperature, and homes with piped water and made of cement. These results indicate that a climate-driven model with limited calibration can capture important epidemic characteristics, which can help guide intervention efforts and improve disease projections associated with climate change.

## Results
**Study sites**. We selected study sites within each country that are distributed across a gradient of temperature, humidity, and rainfall and are climatologically and socioeconomically distinct (Fig. 1 and Table 1). The climate gradient is driven by factors like elevation, distance to the ocean, and the land cover type, as all

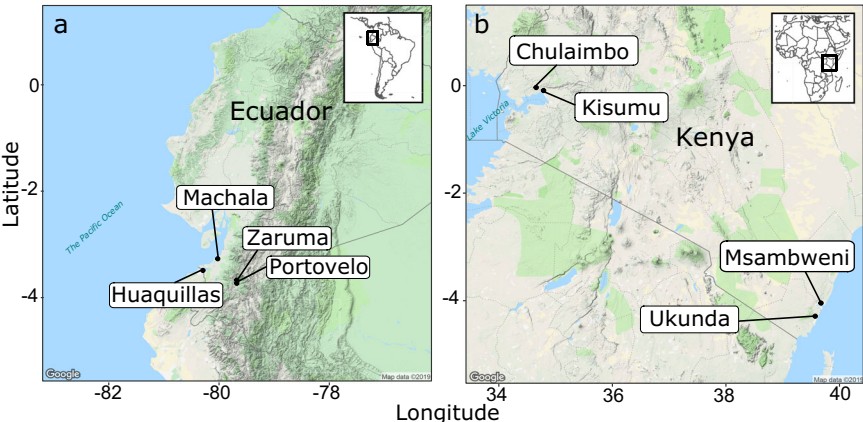

**Fig. 1 Map of study sites. a** Ecuador in South America and **b** Kenya in East Africa. We created this map in R using Google Maps as the base layer.

study sites are located at similar latitudes near the equator. Demographics, housing quality, exposure, susceptibility, and adaptive capacity vary most strongly between the two regions, although there are some differences among sites within the same country.

**Capturing key epidemic characteristics.** The dynamic susceptible, exposed, infectious-susceptible, exposed, infectious, removed (SEI-SEIR) compartmental model parameterized with temperature-, humidity-, and rainfall-dependent mosquito life-history traits (Fig. 2) reproduced three key characteristics of epidemics: number of outbreaks, the timing of outbreak peak, and duration of outbreaks. We defined an outbreak as a continuous-time period with peak cases exceeding the mean number of cases (predicted or observed) plus one standard deviation within a site. Across all sites, the number of outbreaks predicted by the model closely matched the number of outbreaks observed ($R^2 = 0.79$, $P < 0.01$; Fig. 3a). Supporting our a priori expectations based on a previous simulation study[30], we found that the climate-driven model predicted peak timing of outbreaks ($R^2 = 0.71$, $P < 0.01$; Fig. 3b) and outbreak duration ($R^2 = 0.51$, $P < 0.01$; Fig. 3c) well but did not predict the final outbreak size (Fig. 3d) or the maximum number of infections (Fig. 3e) across sites. Overall, the model predicted four outbreaks that were not observed and did not predict five outbreaks that occurred. The model may miss an outbreak (i.e., false negatives) when, for example, a suitable climate occurs but the pathogen is not introduced or the susceptible population is depleted from previous outbreaks.

**Capturing spatiotemporal disease dynamics across sites.** The SEI-SEIR model generated mosquito and disease dynamics that better reflected observed dynamics in some sites than others (Fig. 4 and Table 2). Model-predicted mosquito abundances were significantly correlated with field-collected observations of mosquito abundances in all eight study sites, explaining 28–85% of site-level variation through time based on pairwise correlations with an adjusted $P$ value for time series data (following ref. [43]). Based on surveys conducted across all vector life stages in Kenya (only adult mosquitoes were collected in the Ecuador surveys), the SEI-SEIR model explained variation in the abundance of adult mosquitoes (28–63%) better than pupae (25–32%), late instars (30–33%), early instars (20–36%), and eggs (33–55%), likely because the model did not explicitly incorporate other mosquito life-history stages. Model-predicted disease cases were significantly correlated with the laboratory-confirmed arboviral incidence in seven of the eight study sites, explaining 44–88% of site-level variation through time (within sites with statistically

significant pairwise correlations). We confirmed that the predicted dynamics were stable with sensitivity analyses to initial conditions (see "Methods"), as emerging diseases can display chaotic dynamics due to a high sensitivity to initial conditions. Overall, the model reproduced disease dynamics slightly better for sites in Ecuador compared with Kenya.

We found evidence that rainfall affects transmission through multiple mechanisms and at different time lags (Table 2). Since the effect of rainfall on mosquito abundances is not well-understood, we simulated disease dynamics for each site three times, using one of three hypothesized rainfall relationships (Brière, inverse, and quadratic; Supplementary Fig. 3). We determined the best rainfall function and time lag for each site based on the highest pairwise correlation value between model predictions and observations. The model with the exponentially decreasing inverse rain function (Supplementary Fig. 3c), which indicates that mosquito abundances peak when there is no or low rainfall (likely as a result of water-storage practices and/or unreliable water sources) described observed mosquito and disease dynamics most often, especially in the Kenya sites (Table 2), where household access to piped water is very low (Table 1). The left-skewed unimodal Brière rainfall function (Supplementary Fig. 3a), which indicates that mosquito abundances increase with increasing rainfall until some threshold where flushing occurs, described disease dynamics in some settings, particularly in the Ecuador sites. The symmetric unimodal quadratic rainfall function (Supplementary Fig. 3b), which indicates that mosquito abundances peak with intermediate amounts of rainfall and are reduced with low and high rainfall values, also described disease dynamics in some settings. Interestingly, we did not find a single rainfall function that consistently described dynamics for mosquitoes or arboviral cases across study sites, or for both mosquitoes and arboviral cases within individual study sites (Table 2). In contrast, we did find some consistency with time lags. The model best-predicted mosquito abundances in the same month or 1 month in the future. In more than half of the sites, the model best-predicted human disease cases 3–4 months in the future, and in almost all sites at least 2 months in the future (the exception is Zaruma, where very few arbovirus cases were reported during the study period and were likely due to importation rather than local transmission). Given that multiple rainfall functions and time lags are supported by field data (even within the same study site), we propose a conceptual model that incorporates multiple pathways for rainfall to affect disease dynamics along a continuum of rainfall (Fig. 5), in contrast to distinct functional relationships for a given setting, which motivated the approach used in this study.

**Table 1 Study sites differ geographically, climatologically, and socioeconomically.**

| | Huaquillas, Ecuador | Machala, Ecuador | Portovelo, Ecuador | Zaruma, Ecuador | Chulaimbo, Kenya | Kisumu, Kenya | Msambweni, Kenya | Ukunda, Kenya |
|---|---|---|---|---|---|---|---|---|
| *Site characteristics* | | | | | | | | |
| Elevation (m) | 15 | 6 | 645 | 1155 | 1328 | 1100 | 4 | 8 |
| Location | Coastal | Coastal | Inland | Inland | Inland | Inland | Coastal | Coastal |
| Mean annual NDVI[a] | 0.22 | 0.12 | 0.61 | 0.57 | 0.63 | 0.35 | 0.33 | 0.52 |
| Dominant land cover type[b] | 13 | 13 | 9 | 10 | 14 | 13 | 13 | 10 |
| *Climate* | | | | | | | | |
| Mean temperature (°C) | 26 | 26 | 25 | 22 | 24 | 26 | 28 | 28 |
| Mean relative humidity (%) | 81 | 84 | 81 | 86 | 69 | 50 | 76 | 78 |
| Mean annual rainfall (mm) | 317 | 669 | 500 | 1115 | 1125 | 810 | 1048 | 922 |
| *Demographics* | | | | | | | | |
| Human population size | 57,366 | 279,887 | 13,673 | 25,615 | 7304 | 491,893 | 15,371 | 80,193 |
| Population <5 years (%) | 10 | 9 | 9 | 8 | 12 | 12 | 13 | 14 |
| Population of African ancestry (%) | 5.1 | 6.0 | 3.3 | 2.9 | 100.0 | 100.0 | 100.0 | 100.0 |
| *Housing quality (% houses)* | | | | | | | | |
| Piped water inside home | 90 | 91 | 100 | 96 | 2 | 4 | 3 | 11 |
| No screens on windows | 7 | 60 | 91 | 99 | 74 | 78 | 43 | 21 |
| House materials (cement/mud/wood) | 87/5/0 | 87/8/5 | 95/0/5 | 93/1/1 | 29/70/0 | 77/17/0 | 38/62/0 | 51/47/0 |
| *Exposure, vulnerability, and adaptive capacity* | | | | | | | | |
| Arboviruses present | dengue, chikungunya, Zika | | | | >200 documented including dengue, chikungunya, Yellow fever, Rift Valley fever, West Nile fever, O'nyong-nyong | | | |
| Insecticide use (% houses) | 19 | 28 | 46 | 37 | 0 | 0 | 11 | 55 |
| Bednet use (% houses) | 77 | 55 | 15 | 21 | 93 | 92 | 0 | 96 |
| Other vector control strategies used | Ultra-low volume fumigation with malathion (organophosphate) and community mobilization to eliminate larval habitats | | | | Mosquito coils | | | |
| Annual gross domestic product by country (2018) | $177 billion USD | | | | $85.98 billion USD | | | |

aMean annual normalized difference vegetation index (NDVI) is a proxy for photosynthesis and measured as a difference in spectral reflectance in the visible and near-infrared regions from NASA/NOAA MODIS (MOD13A1)[78].
bDominant land cover type is measured and classified from spectral and temporal features from NASA/NOAA MODIS (MCD12Q1)[79]. Land cover types include (9) tree cover 10–30%, (10) dominated by herbaceous annuals, (13) >30% impervious surface area, and (14) 40–60% mosaics of small-scale cultivation. Bednet use represents the availability of and/or willingness to adopt intervention strategies for preventing infection rather than a direct adaptive response to preventing infection by day-biting *Ae. aegypti* mosquitoes.

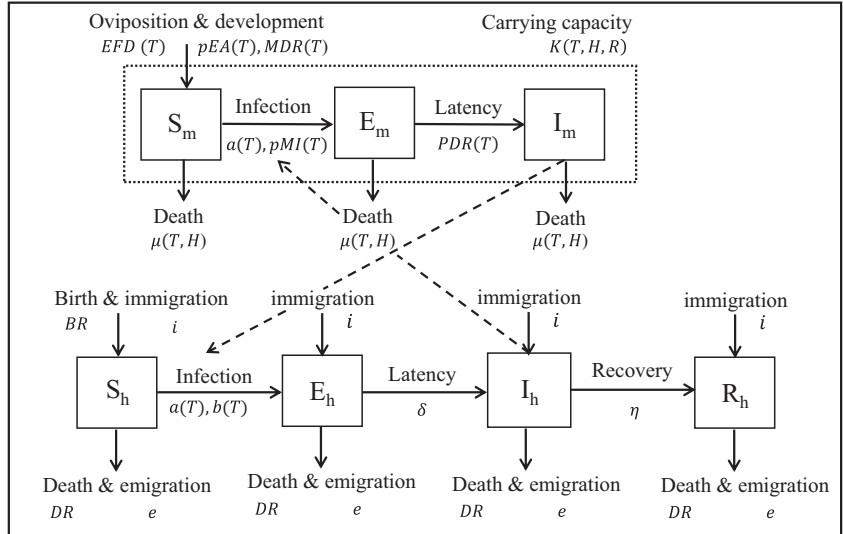

**Fig. 2 SEI-SEIR epidemiological model framework.** The mosquito population is split among susceptible ($S_m$), exposed ($E_m$), and infectious ($I_m$) compartments (squares), and the human population is split among susceptible ($S_h$), exposed ($E_h$), infectious ($I_h$), and recovered ($R_h$) compartments. Solid arrows indicate the direction individuals can move between compartments, and dashed arrows indicate the direction of transmission. Transitions among compartments are labeled by the appropriate processes and corresponding rate parameters (see "Methods" for parameter definitions and more detail). Rate parameters with a $T$, $H$, and $R$ are temperature-, humidity-, and rainfall-dependent, respectively. The total adult mosquito population ($S_m$, $E_m$, and $I_m$ compartments; dotted rectangle) is maintained at an abundance less than or equal to the mosquito-carrying capacity.

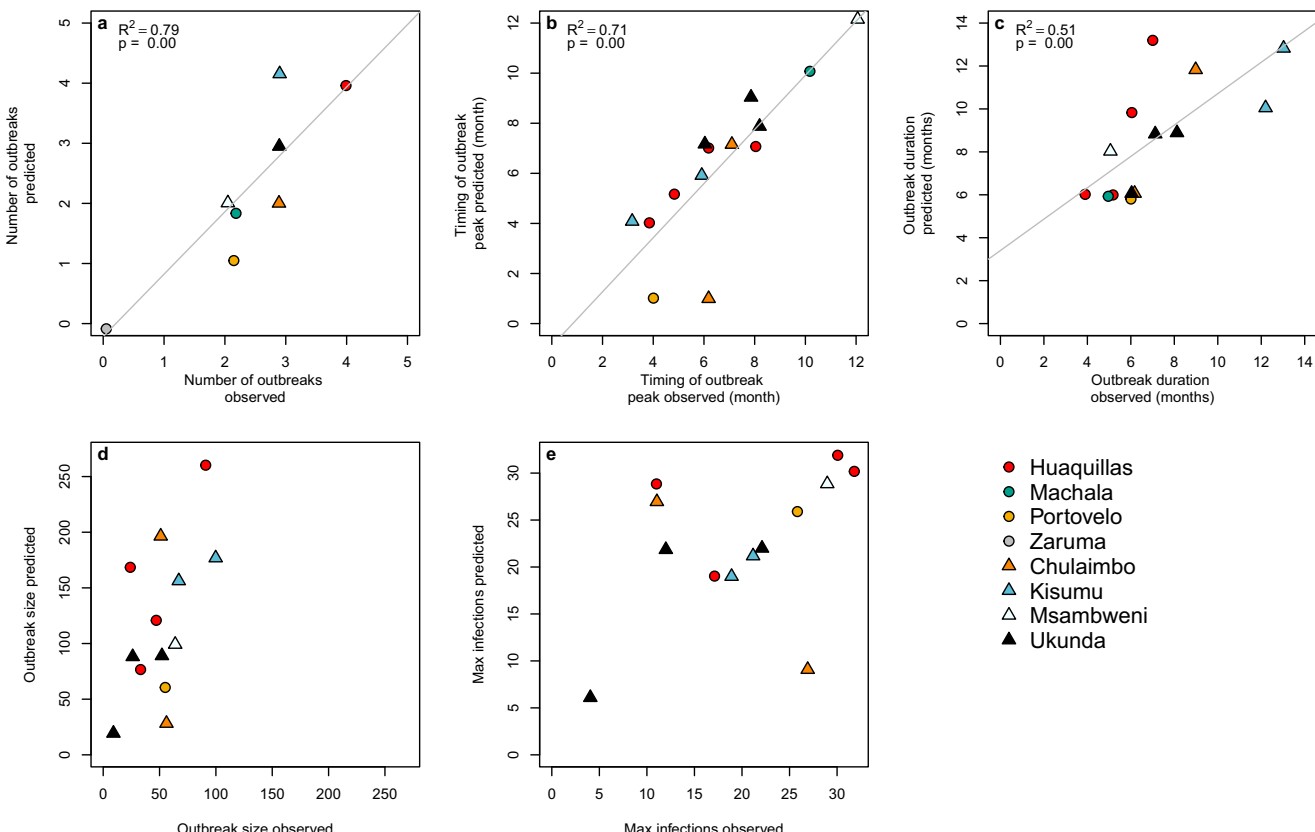

**Fig. 3 Model predictions for the number, timing, and duration of arboviral outbreaks closely matched field observations.** Scatterplots show model predictions versus observations for different epidemic characteristics. **a** The number of outbreaks indicates the total number of predicted and observed outbreaks in a site over the study period. **b** Timing of outbreak peak, **c** outbreak duration, **d** outbreak size, and **e** maximum infections (e.g., max $I_h$ during an outbreak) correspond to individual outbreaks where model predictions and observations overlapped in time, therefore, some plots show multiple data points per site. Outbreaks are colored by site with different symbols for Ecuador (circles) and Kenya (triangles). We show regression lines and associated statistics ($R^2$ = coefficient of determination; $P$ value = probability of two-sided hypothesis test) for statistically significant relationships. For visualization purposes, we jittered the data points to show overlapping data, and we excluded data from Machala in plots (**d**) outbreak size and (**e**) maximum infections because the magnitude differed substantially from all other sites.

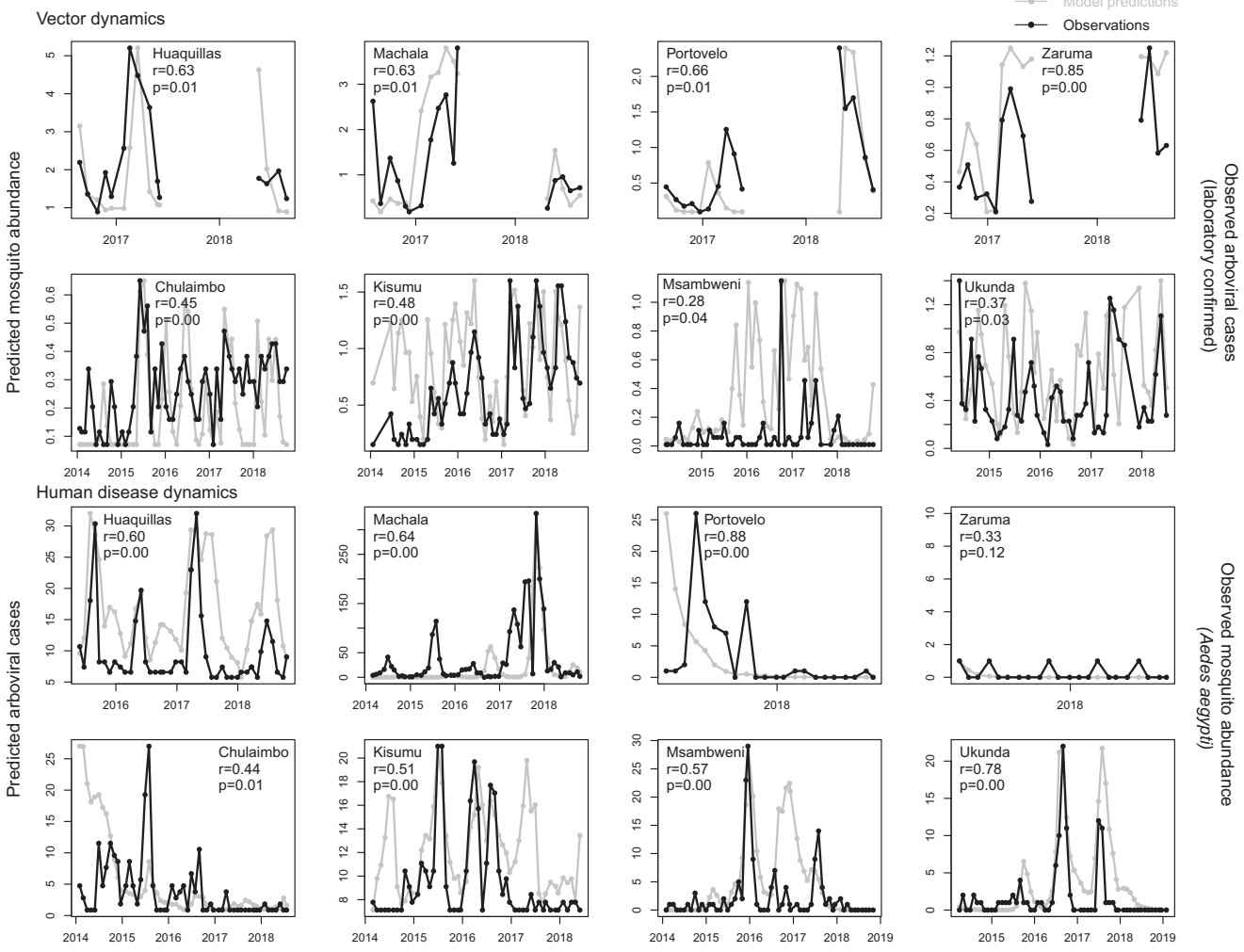

**Fig. 4 Model predicts vector and human disease dynamics better in some settings than others.** Each plot shows the time series of SEI-SEIR model predictions (gray dots connected by gray lines) and field observations (black dots connected by black lines) for vector (top two rows) and human disease (bottom two rows) dynamics for each study site with the pairwise correlation (*r*) and adjusted *P* value for two-sided hypothesis test (*P*). We calculated observed mosquito abundances as the mean number of adult *Ae. aegypti* per house, month, year, and site. We calculated observed arboviral cases as the total number of laboratory-confirmed dengue (any serotype), chikungunya, and Zika cases per month, year, and site; six of the eight study sites only included dengue cases (see "Methods"). The first and third rows show sites in Ecuador, and the second and fourth rows show sites in Kenya. We show uncertainty in model predictions in Supplementary Figs. 1 and 2.

**Table 2 Model predictions reflect a range of observed transmission dynamics when incorporating different rainfall functions and time lags across sites.**

| Site | Vector dynamics | | | | Human disease dynamics | | | |
|---|---|---|---|---|---|---|---|---|
| | Rainfall function | r | Adjusted P value | Lag (months) | Rainfall function | r | Adjusted P value | Lag (months) |
| Huaquillas, Ecuador | Quadratic | 0.63 | 0.01 | 1 | Inverse | 0.60 | 0.00 | 2 |
| Machala, Ecuador | Quadratic | 0.63 | 0.01 | 0 | Brière | 0.64 | 0.00 | 4 |
| Portovelo, Ecuador | Brière | 0.66 | 0.01 | 1 | Brière | 0.88 | 0.00 | 3 |
| Zaruma, Ecuador | Inverse | 0.85 | 0.00 | 1 | Inverse | 0.33 | 0.12 | 0 |
| Chulaimbo, Kenya | Inverse | 0.45 | 0.00 | 1 | Quadratic | 0.44 | 0.01 | 4 |
| Kisumu, Kenya | Brière | 0.48 | 0.00 | 0 | Quadratic | 0.51 | 0.00 | 4 |
| Msambweni, Kenya | Inverse | 0.28 | 0.04 | 0 | Inverse | 0.57 | 0.00 | 3 |
| Ukunda, Kenya | Inverse | 0.37 | 0.03 | 1 | Inverse | 0.78 | 0.00 | 5 |

For each study site, we calculated pairwise correlations between time series of field observations (*Ae. aegypti* abundances or arboviral cases) and time series of model predictions for the SEI-SEIR model with one of three rain functions for mosquito-carrying capacity (Brière, Inverse, or Quadratic) and six-time lags (0–5 months). This table shows specifications for the model (e.g., rain function and time lag) with the highest pairwise correlation value, *r*, for each study site and observation type (vectors or human disease cases), as well as the statistical significance of the correlation value (adjusted *P* value for two-sided hypothesis test) based on the Modified Chelton method[43] to account for temporal autocorrelation.

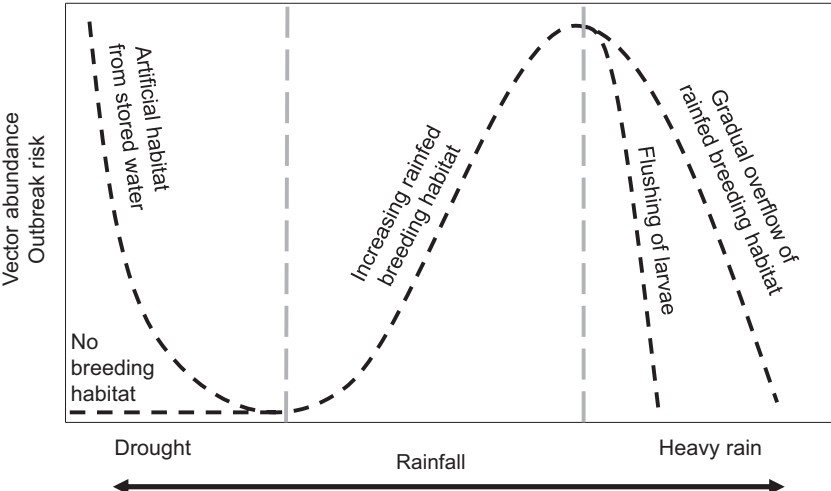

**Fig. 5 Conceptual model for nonlinear functional relationships between rainfall and vector abundance and arboviral outbreak risk.** Dashed lines show multiple potential pathways for rainfall to affect transmission dynamics and include the functional relationships supported in this study. Labels indicate the hypothesized mechanisms along a gradient of rainfall. Adapted from the following source[67].

**Factors that mediate disease dynamics predictability**. The ability of the model to generate similar dynamics to those found in the field varied with demography, housing quality, and climate. Although the sample size is small ($N = 8$ sites), we found that the SEI-SEIR model generally predicted vector dynamics better in sites with a smaller proportion of young children in the population ($R^2 = 0.89$, $P < 0.01$; Fig. 6a), lower mean temperature ($R^2 = 0.63$, $P < 0.05$; Fig. 6c), and a larger proportion of homes with piped water ($R^2 = 0.76$, $P < 0.01$; Fig. 6b) and made of cement ($R^2 = 0.69$, $P < 0.05$; Fig. 6d; list of all factors we assessed are provided in Table 1). Based on the range of mean temperatures at our study sites (22–28 °C), our findings indicate that vector dynamics become less predictable as temperatures near the optimal temperature for transmission (derived in previous studies as 29 °C) following the shape and slope in the $R_0$ curve (Fig. 7). This complements phenomenological models that have found minimal effects of temperature near the empirically derived thermal optima (Fig. 7). None of the socioeconomic factors that we examined in this study (Table 1) explained variability in the pairwise correlations for human disease cases among sites.

## Discussion

Directly observing the influence of climate on species interactions and population dynamics is often challenging because of interacting and nonlinear relationships. Here, we directly and quantitatively connect laboratory-based climate relationships to observed mosquito and disease dynamics in the field, supporting the mechanistic role of climate in these disease systems. The trait-based modeling approach captured several key epidemic characteristics and generated a range of disease dynamics along a spectrum of settings with low levels of transmission to seasonal outbreaks, helping to reconcile seemingly context-dependent effects (i.e., opposite conclusions about the magnitude and direction of effects; Fig. 7) of climate on arboviral transmission dynamics from the literature[7–12,44].

The results of this study shed some light on the influence of climate in driving endemic versus epidemic dengue transmission. Although Ecuador typically experiences seasonal epidemics[6] and Kenya typically experiences low levels of year-round transmission[5], the sites within this study suggest that epidemic transmission is more common in settings with clear seasonality (e.g., coastal sites) whereas endemic transmission is more common in settings with more climate variability (e.g., inland sites),

regardless of country. Coastal sites experienced more regular seasonal climate cycles, likely because oceans buffer climate variability, and this seasonality corresponded with seasonal epidemics. In contrast, the inland sites experienced more day-to-day climate variability, which resulted in more fluctuations in disease cases. As a result, the occurrence and persistence of suitable temperature, rainfall, and humidity conditions enabling outbreaks were less regular in sites with more climate variability. The ability of the model to detect key epidemic characteristics across endemic and epidemic settings indicates that climate plays a major role in driving when outbreaks occur and how long they last.

Using field data on mosquitoes and disease cases from diverse settings and a model parameterized with data from other studies, we identified several key epidemic characteristics that we should (and should not) expect to capture in new settings. While we would never expect a perfect correlation between model predictions and observations, even if the model perfectly captured climate–host–vector dynamics because of the many additional factors that affect transmission in nature, our results indicate that a model with limited calibration can determine the number of outbreaks across settings remarkably well (Fig. 3a). This finding could be particularly useful for prioritizing surveillance or intervention activities across a range of potential sites that would otherwise appear equal in their propensity for outbreaks (e.g., similar climate conditions). We also show that the model captures the peak timing of outbreaks (Fig. 3b) and outbreak duration (Fig. 3c) but not the final outbreak size (Fig. 3d) or the maximum number of infections (Fig. 3e), supporting the hypothesis that the magnitude of disease cases during an outbreak in settings with year-round climate suitability for disease transmission are invariant to temperature, as proposed by[30], likely because the magnitude of disease cases is probably more strongly driven by the availability of susceptible hosts.

Given that the model generally did not predict the magnitude of outbreaks, we asked how well the model reproduced vector and human disease dynamics (i.e., variation over time) across sites and whether this relationship varied systematically with different socioeconomic factors. Across sites, the range of temporal correlations between model predictions and observations ($N = 8$; Fig. 4, Table 2) provides an informative metric for the proportion of true disease dynamics that we might expect to capture in new settings, ranging from 28–88%. The correlations varied with demography, housing construction, and climate (Fig. 6). The

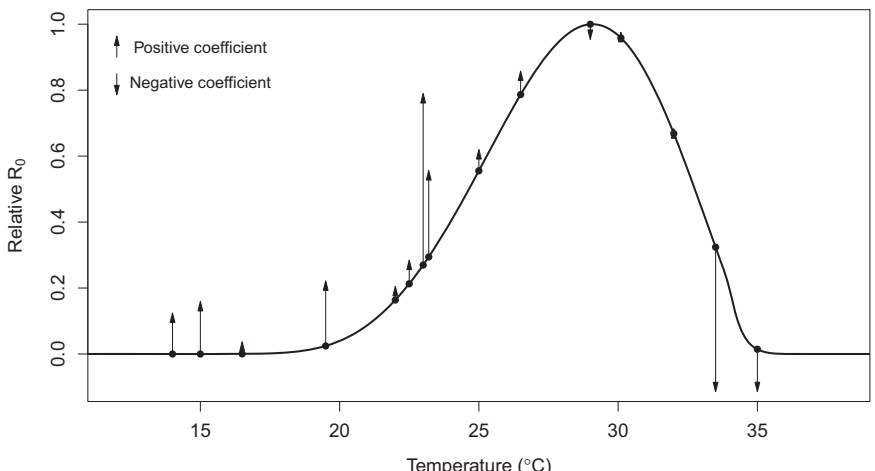

**Fig. 6 Demography, housing construction, and climate affect model predictive capacity for vectors.** Factors that influence the predictability of vector dynamics include (**a**) proportion of the population under five years of age, (**b**) proportion of houses with piped water, (**c**) mean temperature and (**d**) proportion of houses made with cement (walls and/or floors). Points indicate the pairwise correlation value for a single site (colors) with different symbols for Ecuador (circles) and Kenya (triangles). Each plot also shows the linear regression lines and associated statistics ($R^2$ = coefficient of determination; $P$ value = probability of two-sided hypothesis test).

**Fig. 7 Independently predicted relative $R_0$ from a model derived from laboratory studies explains differences in the magnitude and direction of the effects of temperature on dengue transmission in the field across varied settings from previous studies.** The black line shows the relative basic reproductive number ($R_0$, normalized to a 0–1 scale) plotted against temperature based on all temperature-dependent traits from[19] used in the SEI-SEIR model presented here. Points indicate mean temperature values from previous field-based statistical analyses that related dengue cases with the minimum, maximum, or mean ambient temperature; arrows correspond to the direction (up = positive, down = negative) and relative effect size of the temperature–dengue relationship based on coefficient values from the following studies[44,49,68–77]. See "Methods" and Supplementary Table 1 for more detail. As expected, the largest observed positive effects of temperature occurred in the rapidly increasing portion of the $R_0$ curve (~22–25 °C; consistent with findings in this study) and the largest observed negative effects occurred well above the predicted optimum, near the upper thermal limit (~33–35 °C).

model may have better-explained vector dynamics in locations with a lower proportion of children under 5-year old for a variety of reasons, including because bottom-heavy demographic pyramids are often associated with lower socioeconomic status and higher mobility throughout the day. In addition to the demographic makeup of sites, housing construction within sites also seems to modify transmission dynamics: vector dynamics were less predictable in sites with more houses with piped water and made of cement (Fig. 6b, d). These results suggest that piped water may prevent additional contact between humans and mosquitoes associated with stored water around the home. In addition, housing materials like cement that lower indoor temperature could artificially decrease climate suitability for mosquitoes, thereby decreasing the probability that mosquitoes will enter and bite people inside their homes. Despite incorporating all known temperature-dependent mosquito traits into the SEI-SEIR model, we still found vector dynamics became less predictable near the empirically derived thermal optima for arboviral transmission (Figs. 6c, 7). This finding may be associated with physiological or behavioral responses of mosquitoes to temperatures near their thermal safety margin[45,46] and/or humans modifying their environment (as described above) in locations optimal for transmission.

Across the study sites, we found support for three hypothesized relationships between rainfall and mosquito-carrying capacity as well as several time lags between model predictions and disease observations. Support for multiple rainfall functions could indicate that the effects of rainfall on immature habitat are highly heterogeneous, which has been found in previous research in Ecuador[27] and Kenya[47]. Alternatively, the combination of multiple rainfall relationships and time lags could arise from nonlinear and delayed effects of extreme climate such as droughts and floods. More specifically, we hypothesize that there may be multiple mechanistic relationships for the effects of rainfall on mosquito abundance and arboviral disease dynamics (Fig. 5), and they may act on different time scales. For example, previous research indicated that dengue outbreaks were more likely to occur 4–5 months after a drought and 1 month after excessive rainfall and a statistical model that incorporated these duel exposure-lag-response functions was highly effective at predicting dengue outbreaks in Barbados[48]. Further, if multiple rainfall relationships act in concert across varying time lags, this would help to explain why many different time lags have been observed between rainfall and arboviral dynamics in previous studies[6,27,49–52].

Future research can build on this study to improve our understanding of arboviral dynamics across settings. There were several factors that we did not include in this study, such as existing vector control programs, infrastructure, and preexisting immunity in the population. For instance, in Ecuador, factors such as distance to abandoned properties, interruptions in access to piped water, shaded patios, and use of vector control are documented to influence arbovirus transmission[53], whereas in the study sites in Kenya, factors associated with arboviral transmission are less well-studied and there are currently no widely used vector control or local arboviral surveillance programs employed. Future studies could further improve the model by incorporating human immune dynamics associated with interactions among different dengue serotypes[54] or cross-reactivity among viral antibodies[55], differential susceptibility across human age classes[56], and heterogeneity in contact rates between mosquitoes and people based on human behavior and movement[57,58]. Further, as experimental data becomes available for trait estimates specific to chikungunya and Zika, this model could be partitioned to model each arboviral disease individually. This is likely to be an important addition as the different arboviruses tend to peak in

different years, possibly due to differences in viral-development rates and extrinsic incubation periods among arboviruses. Therefore, validating the model with all three arboviruses combined may oversimplify the complex interannual dynamics that arise due to competition among arboviruses in mosquitoes and humans. There were not enough data for chikungunya and Zika cases in this study to formally test such patterns. This study provides strong evidence that a trait-based model, parameterized independently from field data, can reproduce key epidemic characteristics and a range of spatiotemporal arboviral disease dynamics. Such mechanistic, climate-driven models will become increasingly important to support public health efforts in the face of novel climate regimes emerging due to climate change.

## Methods

**Climate data**. We collected in situ measurements of daily mean temperature, relative humidity, and rainfall at each study site and interpolated missing data where necessary. We used temperature and humidity measurements from HOBO loggers and rainfall measurements from rain gauges for sites in Kenya. We used temperature, humidity, and rainfall measurements from automatic weather stations operated by the National Institute of Meteorology and Hydrology in Ecuador. For Kenya, we interpolated missing temperature data from NOAA Global Surface Summary of the Day (Supplementary Table 2 and Supplementary Fig. 4) and interpolated missing rainfall data from NOAA Climate Prediction Center Africa Rainfall Climatology dataset (Supplementary Table 2 and Supplementary Fig. 5). For Ecuador, we interpolated missing temperature (Supplementary Table 2 and Supplementary Fig. 4) and rainfall (Supplementary Table 2 and Supplementary Fig. 5) data using the nearest study site where possible and otherwise based on long-term mean values for the corresponding Julian day. To interpolate missing data, we linearly regressed all measurements taken on the same day in two datasets and then used the linear model to interpolate temperature for the site with missing data based on the climate measurement from the secondary source for the date when the data were missing (Supplementary Figs. 4 and 5). For rainfall, we first calculated a moving window of 14-day accumulated rainfall (which is short enough to capture variability and seasonality in rainfall patterns and follows[59]) for each day before interpolation because modeled daily rainfall values are less reliable than accumulated rainfall over a two week period. We interpolated 14-day cumulative rainfall for any day with a missing rainfall value in the prior 14 days. For both Kenya and Ecuador, we interpolated missing relative humidity data based on long-term mean values for the corresponding Julian day (Supplementary Table 2). We then calculated the saturation vapor pressure deficit (SVPD) from temperature and humidity to use in the humidity function because previous research suggests SVPD is a more informative measure of the effect of humidity on mosquito survival compared with relative humidity[60]. To calculate SVPD, we first calculated the saturation vapor pressure as:

$$SVP = 610.7 * 10^{7.5*T/(273.3+T)} \tag{1}$$

where ($T$) is the temperature in degrees Celsius. We then calculated SVPD (in kilopascals) as

$$SVPD = 1 - \frac{RH}{100} * SVP \tag{2}$$

where RH is relative humidity. The final dataset had no missing values for temperature (Supplementary Fig. 6), rainfall (Supplementary Fig. 7), and humidity (Supplementary Fig. 8).

**Vector surveys**. We collected, counted, sexed, and classified mosquitoes by species, and aggregated the data to the mean number of *Aedes aegypti* per house, month, year, and site to account for differences in survey effort across months and sites. We collected adult mosquitoes using Prokopack aspirators[61]. In Ecuador, we collected mosquitoes from ~27 houses per site (range = 3–57 houses across four sites) every 1–2 weeks during 3, 4-month sampling periods between July 2016 and August 2018 (≈37 sampling weeks per site) to capture different parts of the transmission season. We aggregated the Ecuador vector data to monthly values (≈15 sampling months per site) to correspond with the temporal resolution of surveys in Kenya. In Kenya, we collected mosquitoes from approximately 20 houses per site (range = 1–47 houses across four sites) every month between January 2014 and October 2018 (≈54 sampling months per site). In Kenya, we also collected pupae, late instars, and early instars from containers with standing water around the home and collected eggs by setting ovitraps for an average of four days in and around each house monthly. We brought pupae, late and early instars, and eggs to the insectary and reared them to adulthood to classify individuals by sex and species. All mosquito traps capture a small portion of the true mosquito population; therefore, using consistent trapping methods at the same locations through time allows us to compare relative mosquito population dynamics across study sites rather than the absolute magnitude of mosquito abundances.

**Arboviral surveys**. For Ecuador, we analyzed laboratory-confirmed dengue, chikungunya, and Zika cases provided by the Ministry of Health (MoH) of Ecuador. The MoH collects serum samples from a subset of people with suspected arbovirus infections, and samples are tested at the National Public Health Research Institute by molecular diagnostics (RT-PCR) or antibody tests (IgM ELISA for dengue), depending on the number of days of illness. Results are sent to the MoH Epidemiological Surveillance and Control National Directorate (SIVE Alerta system). Laboratory-confirmed dengue cases were available for all four sites from 2014 to 2018. Laboratory-confirmed chikungunya cases were available for Machala and Huaquillas from 2015 to 2018. Laboratory-confirmed Zika cases were available for Machala from 2016 to 2018.

For Kenya, we used laboratory-confirmed dengue cases aggregated by site and month between 2014 and 2018 collected in a passive surveillance study on childhood febrile illness in Kenya (NIH R01AI102918, PI: ADL). The study population consisted of 7653 children <18 years of age with undifferentiated febrile illness. Children with fever enrolled in the study when attending outpatient care in one of the four study sites (Mbaka Oromo Health Centre in Chulaimbo, Obama Children's Hospital in Kisumu, Msambweni District Hospital in Msambweni, and Ukunda/Diani Health Center in Ukunda). Local health officers collected comprehensive clinical and demographic data and phlebotomy at the initial visit. We tested each child's blood for dengue viremia by molecular diagnostics (conventional PCR[62] or targeted multiplexed real-time PCR when available[63]), or serologic conversion between an initial and a follow-up visit (IgG ELISA[64]).

For arboviral data collection in Ecuador and Kenya, participants provided consent and all local and institutional protocols were followed (Stanford IRB #31488, KEMRI ERC #2611).

**SEI-SEIR model**. We adapted an SEI-SEIR model parameterized for dengue transmission in *Ae. aegypti* mosquitoes[30] to simulate mosquito abundance and arboviral cases through time based on daily weather conditions in eight study locations. The model (Eqs. (3)–(9); Fig. 2), created independently from the observed data described above, allows mosquito life-history traits and viral-development rate to vary with temperature ($T$) following[30], a mosquito-carrying capacity to vary with accumulated 14-day rainfall ($R$) following[59], and mosquito mortality to vary with humidity (i.e., saturation vapor pressure deficit) ($H$) following[60].

$$\frac{dS_m}{dt} = \varphi(T,H) * \frac{1}{\mu(T,H)} * N_m * \left(1 - \frac{N_m}{K(T,R,H)}\right) - \left(a(T) * pMI(T) * \frac{I_h}{N_h} + \mu(T,H)\right) * S_m \quad (3)$$

$$\frac{dE_m}{dt} = a(T) * pMI(T) * \frac{I_h}{N_h} * S_m - (PDR(T) + \mu(T,H)) * E_m \quad (4)$$

$$\frac{dI_m}{dt} = PDR(T) * E_m - \mu(T,H) * I_m \quad (5)$$

$$\frac{dS_h}{dt} = -a(T) * b(T) * \frac{I_m}{N_h} * S_h + BR * S_h - DR * S_h + ie * N_h - ie * S_h \quad (6)$$

$$\frac{dE_h}{dt} = a(T) * b(T) * \frac{I_m}{N_h} * S_h - \delta * E_h - DR * E_h - ie * E_h \quad (7)$$

$$\frac{dI_h}{dt} = \delta * E_h - \eta * I_h - DR * I_h - ie * I_h \quad (8)$$

$$\frac{dR_h}{dt} = \eta * I_h - DR * R_h - ie * R_h \quad (9)$$

where

$$\varphi(T,H) = EFD(T) * pEA(T) * MDR(T) \quad (10)$$

The adult mosquito population ($N_m$) is separated into susceptible ($S_m$), exposed ($E_m$), and infectious ($I_m$) compartments, and the human population ($N_h$) is separated into susceptible ($S_h$), exposed ($E_h$), infectious ($I_h$), and recovered ($R_h$) compartments (Fig. 2). Climate-independent model parameters include the intrinsic incubation period ($\delta = 5.9$ days), human infectivity period ($\eta = -5$ days), birth rate (BR = 31.782 and 20.175 per 1000 people in Ecuador and Kenya, respectively), death rate (DR = 5.284 and 5.121 per 1000 people for Ecuador and Kenya, respectively), and immigration/emigration rate (i.e. = 0.01). The temperature-dependent SEI-SEIR model was developed by Huber et al.[30] and allows mosquito life-history traits and viral-development rate to vary according to thermal response curves fit from data derived in laboratory experiments conducted at constant temperatures (Table 3). Although laboratory experiments do not reflect real-world conditions, the physiological responses measured are biologically meaningful. The temperature-dependent traits include eggs laid per female per day (EFD), the probability of egg-to-adult survival (pEA), mosquito development rate (MDR), mosquito mortality rate (lifespan$^{-1}$; $\mu$), biting rate ($a$), probability of mosquito infection per bite on an infectious host (pMI), parasite development rate (PDR), and the probability of mosquito infectiousness given an infectious bite ($b$).

We modified the mosquito mortality rate equation to vary as a function of temperature and humidity by fitting a spline model based on a pooled survival analysis of *Ae. aegypti*[60] (Supplementary Fig. 9):

$$\mu(T,H) = \frac{1}{c * (T - T_0) * (T - T_m)} + (1 - (0.01 + 2.01 * H)) * y \quad H < 1 \quad (11)$$

$$\mu(T,H) = \frac{1}{c * (T - T_0) * (T - T_m)} + (1 - (1.22 + 0.27 * H)) * y \quad H \geq 1 \quad (12)$$

where the rate constant ($c$), minimum temperature ($T_0$), and maximum temperature ($T_m$) equal $-1.24$, 16.63, and 31.85, respectively (Table 3), humidity ($H$) is the saturation vapor pressure deficit, and $y$ is a scaling factor that we set to 0.005 and 0.01, respectively, to restrict mosquito mortality rates within the range of mortality rates estimated by other studies[19,60]. The linear humidity function has a steeper slope at lower humidity values (Eq. (11)) compared with higher humidity values (Eq. (12)) based on previous research[60] (Supplementary Fig. 9).

We modeled adult mosquito-carrying capacity, $K$, as a modified Arrhenius equation following[30,65]:

$$K(T,H,R) = \frac{EFD(T_0) * pEA(T_0) * MDR(T_0) * \mu(T_0,H_0)^{-1} - \mu(T_0,H_0)}{EFD(T_0) * pEA(T_0) * MDR(T_0) * \mu(T_0,H_0)^{-1}} * N_{m.max} * e^{\frac{-E_A * (T - T_0)^2}{K_B * (T + 273) * (T_0 + 273)}} * f(R) \quad (13)$$

with $T_0$ and $H_0$ set to the temperature and humidity where carrying capacity is greatest (i.e., physiological optimal conditions from laboratory experiments; 29 °C and 6 kPA), $N_{m,max}$ set to the maximum possible mosquito abundance in a population (twice the human population size following[30]), and the Boltzmann constant, ($K_B$), is $8.617 \times 10^{-5}$ eV/K. We set the activation energy, $E_A$, as 0.05 based on ref. [30]. Since there were no experimental data from which to derive the functional response of mosquito-carrying capacity across a gradient of rainfall values, we tested several functional relationships based on hypothesized biological relationships between freshwater availability and immature mosquito breeding habitat, modeling the effect of rainfall on carrying capacity, $f(R)$, as either:

$$f(R_{Briere}) = c * R * (R - R_{min}) * \sqrt{(R_{max} - R)} * z \quad (14)$$

$$f(R_{Quadratic}) = c * (R - R_{min}) * (R - R_{max}) * z \quad (15)$$

$$f(R_{Inverse}) = \frac{1}{R} * z \quad (16)$$

where minimum rainfall ($R_{min}$) equaled 1 mm and maximum rainfall ($R_{max}$) equaled 123 mm based on the high probability of flushing[26]. The quadratic function is similar to the rainfall function found in ref. [26] and the inverse function is based on the rainfall function used in ref. [59]. We used rate constants ($c$) of $7.86e^{-5}$ and $-5.99e^{-3}$ for the Brière and quadratic functions, respectively, based on rate constants for other parameters with similar functional forms (Table 3). We also included a scaling factor, $z$ (0. 28, 0.025, and 0.60, respectively), to restrict the maximum carrying capacity to produce model outputs based on a subsample of the total population for comparison with observations. Since the rate constant, $c$, is multiplied by $z$, inferring the exact value of $c$ is not necessary because it is scaled by $z$. The scaling factor could be removed from the model to simulate dynamics in the total population.

To initiate the model, we used site-specific values for human population size and randomly selected one set of values for all sites for the proportion of mosquitoes and humans in each compartment. For Ecuador, we used population estimates from official population projections produced by Proyección de la Población Ecuatoriana, por años calendario, según cantones 2010–2020 (https://www.ecuadorencifras.gob.ec/proyecciones-poblacionales/) with population sizes of 57,366, 279,887, 13,673, and 25,615 for Huaquillas, Machala, Portovelo, and Zaruma, respectively, based on 2017 projections. For Kenya, we estimated the population sizes served by each outpatient care facility by creating a polygon around all the geolocations of study participants' homes enrolled at each outpatient care facility and summed population count data from NASA's Socioeconomic Data and Applications Center Gridded Population of the World v4 (https://doi.org/10.7927/H4JW8BX5) within each polygon using ArcGIS v 10.4.1. We estimated population sizes of 7,304, 547,557, 240,698, and 154,048 for Chulaimbo, Kisumu, Msambweni, and Ukunda, respectively. We set the ratio of mosquitoes to humans to two, following[30]. We used the following values as the initial proportion of mosquitoes and humans in each model compartment: $S_m = 0.22$, $E_m = 0.29$, $I_m = 0.49$, $S_h = 0.58$, $E_h = 0.22$, $I_h = 0.00$, and $R_h = 0.20$. We determined that the model was invariant to initial proportion values after a short burn-in period (90 days) based on sensitivity analysis (Supplementary Fig. 10); therefore, we randomly selected one set of initial proportion values from the sensitivity analysis for all the model simulations. We also determined that the temporal trajectories of model dynamics did not change when we varied the critical thermal minimum, maximum, and rate constants (Table 3) for *Aedes aegypti* life-history traits (Supplementary Figs. 1 and 2).

We ran all model simulations using the deSolve package in R statistical software v 3.5.3[66].

**Model validation**. To validate the SEI-SEIR model, we calculated pairwise correlations with an adjusted p value to account for autocorrelation for each site. For the

**Table 3 Fitted thermal responses for *Ae. aegypti* life-history traits.**

| Trait | Definition | Function | c | $T_0$ | $T_m$ |
|---|---|---|---|---|---|
| a | Biting rate (day$^{-1}$) | Brière | $2.02e^{-04}$ | 13.35 | 40.08 |
| EFD | Eggs laid per female per day | Brière | $8.56e^{-03}$ | 14.58 | 34.61 |
| pEA | Probability of mosquito egg-to-adult survival | Quadratic | $-5.99e^{-03}$ | 13.56 | 38.29 |
| MDR | Mosquito egg-to-adult development rate (day$^{-1}$) | Brière | $7.86e^{-054}$ | 11.36 | 39.17 |
| Lf | Adult mosquito lifespan (days) | Quadratic | $-1.48e^{-01}$ | 9.16 | 37.73 |
| b | Probability of mosquito infectiousness | Brière | $8.49e^{-04}$ | 17.05 | 35.83 |
| pMI | Probability of mosquito infection | Brière | $4.91e^{-04}$ | 12.22 | 37.46 |
| PDR | Parasite development rate (day$^{-1}$) | Brière | $6.56e^{-05}$ | 10.68 | 45.90 |

Traits were fit to a Brière [$cT(T-T_0)(T_m-T)^{\frac{1}{2}}$] or a quadratic [$c(T-T_m)(T-T_0)$] function where $T$ represents temperature. $T_0$ and $T_m$ are the critical thermal minimum and maximum, respectively, and $c$ is the rate constant. Thermal responses were fit by ref. [19] and also used in ref. [30]. Parasite development rate was measured as the virus extrinsic incubation rate.

pairwise correlations, we used the ccf function in base R[66] to calculate correlations between the two time series of model predictions and observations with 0, 1, 2, 3, 4, and 5-month lags. We then calculated an adjusted p value using the Modified Chelton method[43] to adjust the null hypothesis test of sample correlation between autocorrelated time series. To assess predictions and observations for vector dynamics for each site, we compared the monthly time series of the total predicted mosquito population from the SEI-SEIR model with the monthly time series of the mean number of *Aedes aegypti* (per house). We followed the same procedure to compare model predictions with other mosquito life stages for sites in Kenya. Similarly, to compare predictions and observations for human disease dynamics for each site, we compared the monthly time series of predicted infected individuals from the SEI-SEIR model with the monthly time series of total laboratory-confirmed arboviral cases. For subsequent analyses, we used model predictions from the model (e.g., SEI-SEIR model with a specific rainfall function and time lag) with the highest pairwise correlation value.

To compare key epidemic characteristics between model predictions and observations and to compare site-specific correlations with socioeconomic factors, we used linear regression models using the lm function in the stats package in R[66]. We defined outbreaks as a continuous-time period where the peak cases exceeded the mean number of cases (predicted or observed) plus one standard deviation within a site. We then used those outbreak periods to count the total number of outbreaks within each site, and, for predicted and observed outbreaks that overlapped in time, the duration, peak timing, total outbreak size, and the maximum number of infections. We compared predictions and observations for each of these metrics with linear regression. Since we were interested in whether model predictions matched observations for each independent outbreak period, we did not allow varying intercepts or slopes by site. Similarly, we compared the pairwise correlation values (described above) across all sites with each socioeconomic factor listed in Table 1 separately using linear regressions.

**Comparison of $R_0$ with prior studies**. We collected effect sizes of temperature on dengue incidence from 12 peer-reviewed studies from the literature (Supplementary Table 1). We selected studies with mean temperatures across the predicted temperature range where arboviral transmission can occur. We scaled the coefficient values to visualize the relative effect of temperature across studies given that the original analyses were conducted with different temperature metrics and across different temperature ranges. We provide additional information and sources in Supplementary Table 1.

**Reporting summary**. Further information on research design is available in the Nature Research Reporting Summary linked to this article.

## Data availability

All data that support the findings of this study are deposited in a public repository, with the exception of arboviral incidence data from Ecuador, which is available from the corresponding author upon reasonable request. Climate, epidemic characteristics, socioeconomic, vector, and Kenya arboviral incidence data that we analyzed in this study are available in the following public repository: https://github.com/jms5151/SEI-SEIR_Arboviruses. This data supported Figs. 3, 4, and 6 in the main text and Supplementary Figs. 1–3 and 6–10. Data that support Fig. 7 are available in Supplementary Table 1. The arboviral case data from Ecuador are available from the corresponding author upon reasonable request. The data are not publicly available due to a Confidentiality Agreement with the Ecuador Ministry of Health. Crude birth and death rates used in the model are from The World Bank Open Data. The URL for crude birth rate in Ecuador is https://data.worldbank.org/indicator/SP.DYN.CBRT.IN?locations=EC and for Kenya is https://data.worldbank.org/indicator/SP.DYN.CBRT.IN?locations=KE. The URL for crude death rate for Ecuador is https://data.worldbank.org/indicator/SP.DYN.CDRT.IN?locations=EC and for Kenya is https://data.worldbank.org/indicator/SP.DYN.CDRT.IN?locations=KE.

## Code availability

Model and analysis codes are available at https://github.com/jms5151/SEI-SEIR_Arboviruses.

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

## Acknowledgements

J.M.C., A.D.L., E.F.L., and E.A.M. were supported by a Stanford Woods Institute for the Environment—Environmental Ventures Program grant (PIs: E.A.M., A.D.L., and E.F.L.). E.A.M. was also supported by a Hellman Faculty Fellowship and a Terman Award. A.D.L., B.A.N., F.M.M., E.N.G.S., M.S.S., A.R.K., R.D., A.A., and H.N.N. were supported by a National Institutes of Health R01 grant (AI102918; PI: A.D.L.). E.A.M., A.M.S.I., and S.J.R. were supported by a National Science Foundation (NSF) Ecology and Evolution of Infectious Diseases (EEID) grant (DEB-1518681), and A.M.S.I. and S.J.R. were also supported by an NSF DEB RAPID grant (1641145). E.A.M. was also supported by a National Institute of General Medical Sciences Maximizing Investigators' Research Award grant (R35GM133439) and an NSF and Fogarty International Center EEID grant (DEB-2011147). We thank Cat Lippi for assistance with formatting household quality survey data from Ecuador.

## Author contributions

E.A.M., A.D.L., E.F.L., and J.M.C. conceived of the project. J.M.C. conducted analyses and wrote the paper. E.A.M., A.D.L., E.F.L., and A.M.S.I. secured funding for the project. B.N.N., F.M.M., E.B.A., A.A., M.J.B.C., R.D., F.H.H., R.M., and H.N.N. collected the data. E.N.G.S. and M.M.S. conducted laboratory analyses. A.R.K., S.J.R., and R.S. processed the data. All authors revised and approved the paper.

## Competing interests

The authors declare no competing interests.
