## [Peer Review File · Nature Communications]

Reviewers' comments:

Reviewer #1 (Remarks to the Author):

This manuscript seeks to evaluate the ability of a 'bottom-up' process-based model for the population dynamics of arboviruses to explain vector numbers and reported cases in different locations in Kenya and Ecuador. These locations are chosen along gradients in climate conditions. Furthermore, the authors use the model to evaluate interventions.

The question of whether this kind of model, parameterized independently from the field data, can predict important features of the temporal dynamics and spatial variation in incidence or vector abundance, is an interesting one. In their previous work, the authors have developed these models and demonstrated their usefulness. It is not clear however that these mechanistic models can be built only from 'first principles' (the parameterizations from the lab and literature) without any calibration of at least some parameters to data from a given location of interest. It would be surprising to show that this is possible. I have my doubts but was intrigued by the work.

In particular, the authors motivate the work by saying that such a model should be able to explain contrasting locations in terms of dengue dynamics (and possibly that of other arbovirus diseases). This is an interesting challenge. The other argument that this kind of model can explain contrasting correlative results for climate drivers is less interesting because we do know that the effects of these drivers are nonlinear and that therefore they can have different directions depending on where we are relative to the peak of the reproductive number R_0 . Similarly the conclusion on what part of the vector's life cycle has a stronger impact when intervening, is not completely novel. The early models for vector-borne infections were used to make this point.

I have a series of major concerns outlined below which made the results and their interpretation unconvincing.

1) I expected to see something on the contrasting patterns the model would explain. These could be patterns in the temporal trends, seasonal dynamics, interannual variability, spatial variation with the gradients. The introduction mentions differences in the interannual dynamics for Kenya and Ecuador but we are not shown these in the data used in the paper. This absence of patterns makes the work that follows less clear, especially in its relation to one of the main goals stated in the Introduction.

2) The statement that the model is able to explain the temporal patterns of incidence in different locations is not supported by the results. In particular, Figure 4 is unconvincing. In one of the two locations, the model predicts essentially fairly regular seasonal outbreaks with little variation from year to year. This is not at all the behavior that is shown for the cases in the different diseases. In fact, the cases show intermittent epidemics with seasons way below the predicted incidence. The comparison in the second location is somewhat more convincing. Does the model really capture the main difference outlined in the Introduction for Kenya and Ecuador? We are never shown this.

3) The comparison of predictions and observations for the mosquito abundances and disease incidences based z-scores, was confusing. I understand that one may want to evaluate a discrepancy in terms of some common ruler, and that using a z-score may help in that way. But I do not follow the comparison of the z-score of observations to that of predictions. It is very difficult to get a sense for what we are really comparing here. It also depends strongly on how we choose the variance to obtain a z-score. Which variance and which mean is being used? One could also use a z-score that considers the prediction and how far it is from the mean of the observations, with some variance as a ruler. Does the assumption of a normal distribution makes sense? In any case, the way this is done is poorly justified and difficult to evaluate in terms of what it really tells us.

4) As for (3), I cannot interpret what it means to have a significant correlation in these z-scores. Also, the variance explained in this relationship is quite small (especially for the incidences).

5) I would have liked to see a clearer explanation of how the rainfall enters in the model. Temperature effects have a laboratory basis on now substantial work. Rainfall is what will determine the 'carrying capacity' or overall level of the mosquito population. It is an extremely difficult variable to work with. How is this calibrated independently from location? Is this even possible?

Reviewer #2 (Remarks to the Author):

See attached.

Reviewer #3 (Remarks to the Author):

The manuscript presents an interesting analysis of the mechanistic effect of climate on transmission of arboviral infections. By mechanistically linking temperature-mosquito traits relationships to a dynamic compartmental model of transmission of arboviruses, the authors investigated how well climatic factors such as temperature, humidity and precipitation drives observed patterns of dengue, zika and chikungunya cases in two settings of Kenya and Ecuador. I find the study novel and of interest to the scientific community. I have a few comments which might help strengthening the aims and conclusions of the paper:

1) The introduction to the study stresses on the idiosyncratic nature of climate effects of disease transmission, e.g. lines 69-84. However, I find that the results, apart from the rainfall-mosquito relationship (Table 2), do not emphasize much this aspect, specifically the fact that the effect of climate on arbovirus transmission in Kenya and Ecuador is context-dependent. I would find it interesting to see more results on this. For instance, Figure 4 shows model predictions and observed number of cases for two settings in Ecuador. I would like to see how model predictions compare to observed number of cases in Kenya as well. Furthermore, while a measure of overall fit by settings/sites is reported in the supplementary material, I think it is important for the purpose of applying the results to other settings, to also show how the model predicts observed data at different times throughout the epidemics. I would also like to know a bit more on the climatic conditions leading to the model under predicting and over predicting in different sites in Ecuador and Kenya. It seems that the CART analysis was conducted on the overall dataset. A CART analysis conducted for the different settings would be quite informative on the aspects which make the effect of climate unique across settings. Generally I think it would be useful, and it would help the paper to align to its initial claims regarding the scope of the research, if the authors could highlight a bit more in the results the differential effects of climate across sites and expand on the causes of these differences in the discussion.

2) I am curious to why the model does not seem to predict dengue cases as well as it seem to do with Zika and Chikungunya. From Figure 4 (although it is not clear as observed cases are aggregated across diseases) it looks like the model over predicts dengue cases at quite a few times. Can the author provide some justification to why they think this is the case? Also, it would be useful to see a time-dependent figure of the comparison between observations and model predictions shown for the different viruses.

3) I am a bit unsure regarding the practical applicability of the intervention simulations. To simulate the effect of interventions the authors varied three mosquito traits (mosquito biting rate, carrying capacity and mosquito abundance) by different amounts and recalculated the total number of cases by running forward the compartmental model. Intervention assessment can also be done by varying epidemiologically variables such as the basic reproduction number of the disease, R_0 , which can be reasonably estimated from national disease surveillance system. In this

case, the authors chose to vary mosquito traits, and this in reality would require information on these traits, which might not be available. Some discussion around this issue would be useful. Also it would be fairer to clarify that the results of this intervention modelling exercise (specifically the reduction in burden) are purely based on lab data and simulation and they do not include any data collected in the field (e.g. disease case number). They were not tested or validated. This obviously limits the degree to which they can be trusted or extrapolated to other settings. Providing a bit more context, for instance on why certain effect sizes (10,50,90%) were simulated, how they can be measured (need to have information on those traits in the field) and how they be implemented in practice would help clarifying the relevance of this analysis and strengthen the applicability of its results.

Some minor comments:

- Line 57. It is a bit vague. What are the contradictory results and how this paper helps resolving them?
- Line 59 "tested". I would tone this down a bit – the intervention results come from untested effect sizes and they might be difficult to use in reality due to general lack of field data on mosquito traits in the field.
- Line 91. Another control strategy for Aedes-borne viruses is Wolbachia. Maybe worth mentioning it.
- Line 120. What is the original study (reference) of the model used in this paper?
- Lines 124-138. These distinctions between study settings are interesting. It would be interesting to see in the results a stronger emphasis on the distinct effects of climate on disease transmission across settings. Otherwise at the moment, this section does not seem to have a strong reason to be here.
- Line 151 "Site-months". Are predictions made by months? I find the term unclear.
- Figure 2. Perhaps consider colouring only the boxes rather than whole areas – to highlight the different elements of the model (not just the host and the vector)
- Figure 3. The title of panel b should be "cases"? It would be interesting to see this correlation performed separately for Ecuador and Kenya, if possible.
- Figure 4. What is the fit between model predictions and the observed temporal trends of cases in Kenya?
- Figure 4. Also, can model predictions can be disaggregated by disease? It is difficult to compare them with data shown by disease.
- Line 290. "Predicting" might be more appropriate here than "evaluating".
- Line 323-335. This part sounds a bit like a repetition of the introduction. The authors might consider expanding here on their results and the novelty of the study.
- Line 340-342. I would highlight these important implications in the introduction.
- Line 566. Why were these effect sizes chosen? Is there any indication that any of these values is a realistic target to aim for?

Reviewers' comments:

Reviewer #1 (Remarks to the Author):

This manuscript seeks to evaluate the ability of a 'bottom-up' process-based model for the population dynamics of arboviruses to explain vector numbers and reported cases in different locations in Kenya and Ecuador. These locations are chosen along gradients in climate conditions. Furthermore, the authors use the model to evaluate interventions.

The question of whether this kind of model, parameterized independently from the field data, can predict important features of the temporal dynamics and spatial variation in incidence or vector abundance, is an interesting one. In their previous work, the authors have developed these models and demonstrated their usefulness. It is not clear however that these mechanistic models can be built only from 'first principles' (the parameterizations from the lab and literature) without any calibration of at least some parameters to data from a given location of interest. It would be surprising to show that this is possible. I have my doubts but was intrigued by the work.

In particular, the authors motivate the work by saying that such a model should be able to explain contrasting locations in terms of dengue dynamics (and possibly that of other arbovirus diseases). This is an interesting challenge. The other argument that this kind of model can explain contrasting correlative results for climate drivers is less interesting because we do know that the effects of these drivers are nonlinear and that therefore they can have different directions depending on where we are relative to the peak of the reproductive number R_0 . Similarly the conclusion on what part of the vector's life cycle has a stronger impact when intervening, is not completely novel. The early models for vector-borne infections were used to make this point.

I have a series of major concerns outlined below which made the results and their interpretation unconvincing.

Reviewer # 1 Comment # 1: I expected to see something on the contrasting patterns the model would explain. These could be patterns in the temporal trends, seasonal dynamics, interannual variability, spatial variation with the gradients. The introduction mentions differences in the interannual dynamics for Kenya and Ecuador but we are not shown these in the data used in the paper. This absence of patterns makes the work that follows less clear, especially in its relation to one of the main goals stated in the Introduction.

Response: Thank you for this important comment. We have expanded our explanation of these types of patterns in several ways. (1) Table 1 now presents site-specific characteristics that describe differences in geography, climate, demographics, and vulnerability to mosquito-borne disease transmission. (2) We

show site-specific dynamics of vectors and disease cases—overlaid with the model prediction—in a new figure (Figure 3). (3) We plot the relationships between predictability of vector and human disease dynamics and a variety of site characteristics in a new figure (Figure 4). (4) We discuss these findings in the Results and Discussion.

Reviewer # 1 Comment # 2: The statement that the model is able to explain the temporal patterns of incidence in different locations is not supported by the results. In particular, Figure 4 is unconvincing. In one of the two locations, the model predicts essentially fairly regular seasonal outbreaks with little variation from year to year. This is not at all the behavior that is shown for the cases in the different diseases. In fact, the cases show intermittent epidemics with seasons way below the predicted incidence. The comparison in the second location is somewhat more convincing. Does the model really capture the main difference outlined in the Introduction for Kenya and Ecuador? We are never shown this.

Response: This is an important comment. To address this and similar comments from other reviewers, we have replaced our original Figures 3 and 4 with a figure showing the model predictions and observations separated by site and vector and human disease dynamics (the new Figure 3). This new figure makes it clear that there are a wide range of disease dynamics across sites and that the model generates those dynamics for many sites.

Reviewer # 1 Comment # 3: The comparison of predictions and observations for the mosquito abundances and disease incidences based z-scores, was confusing. I understand that one may want to evaluate a discrepancy in terms of some common ruler, and that using a z-score may help in that way. But I do not follow the comparison of the z-score of observations to that of predictions. It is very difficult to get a sense for what we are really comparing here. It also depends strongly on how we choose the variance to obtain a z-score. Which variance and which mean is being used? One could also use a z-score that considers the prediction and how far it is from the mean of the observations, with some variance as a ruler. Does the assumption of a normal distribution makes sense? In any case, the way this is done is poorly justified and difficult to evaluate in terms of what it really tells us.

Response: Thank you for this important comment. We have revised our statistical method for comparing model predictions with observations. We now calculate pairwise correlations for the time series of predictions and observations for each site separated by vector and human disease dynamics. We further use an adjusted p-value to account for autocorrelation associated with time series data to determine whether the pairwise correlation values are statistically significant. We think this comparison provides a much better indication of model fit and appreciate the reviewer pointing out the issues with the original statistical method we used to assess model fit.

Reviewer # 1 Comment # 4: As for (3), I cannot interpret what it means to have a significant correlation in these z-scores. Also, the variance explained in this relationship is quite small (especially for the incidences).

Response: In response to the above comment, we no longer use z-scores as a metric of model fit in this paper. The new correlation metric has a more straightforward interpretation, indicating the strength and direction of the linear relationship between the model predictions and observations.

Reviewer # 1 Comment # 5: I would have liked to see a clearer explanation of how the rainfall enters in the model. Temperature effects have a laboratory basis on now substantial work. Rainfall is what will determine the ‘carrying capacity’ or overall level of the mosquito population. It is an extremely difficult variable to work with. How is this calibrated independently from location? Is this even possible?

Response: We have expanded the explanation of where rainfall enters the model in the main text and indicate how rainfall enters the model in the conceptual framework (Figure 2). To select the most appropriate rainfall – carrying capacity relationship in each setting, we simulated disease dynamics using three rainfall functions and then selected the model that best reflected observed dynamics. We explain this process in the Results and Methods sections and indicate which rainfall function was used in the selected model for each site in Table 2. We also discuss the interpretation and implications of these functions in the discussion of the new Figure 6.

Reviewer # 2 (Remarks to the Author):

Review of “*Climate explains geographic and temporal variation in mosquito-borne disease dynamics on two continents*” by Caldwell et al. for Nat Comm.

The manuscript develops a life history trait-based model based on a vector-human SEI-SEIR framework to define the climate-driven mechanisms affecting *Aedes aegyptii* population dynamics and arbovirus cases in Equador and Kenya. The model is parameterized using biological values from the literature and climate data from field observations, and is validated with field observations and national records of disease incidence. The authors then use the model to simulate the impact of three different vector control interventions against arbovirus and show that reducing contact rate and immature larval habitats is more effective than reducing adult population size.

Developing a way to understand the mechanisms underlying vector population dynamics is critically important for optimising control of vector-borne diseases, and the modelling framework presented here is promising. However, I have

concerns over the model validation and relevance of the findings. I expand on these and provide other comments below:

Major comments:

Reviewer # 2 Comment # 1: I laud the use of field data to validate a traditionally theoretical model. It is a shame that this is not a more common practice. However, a correspondence of 65% (or for some life history stages 50% and some sites 45%) and average correlation coefficients of 0.35 and 0.19 between predictions and observations seem very low to consider a model appropriate. In addition, the dynamics of the predicted cases as presented in Figure 4 do not seem to follow the pattern or magnitude of the case data, for a single virus nor all viruses combined. Given these values, can we consider the model robust and biologically relevant?

Response: Thank you for this comment. We agree that the original metrics and figures we presented for model fit made it unclear whether we developed a robust model. We now more clearly demonstrate that the model captures important features of the range of disease dynamics observed across sites. In response to this and other reviewer comments, we have re-evaluated how we assess and present the results. We now use pairwise correlation values with an adjusted p-value to quantify model fit and present a new figure (Fig. 3) of site-specific time series plots for all model predictions and observations. The updated model fit values can range from -100 to 100% (perfect fit) where a completely random model should result in 0% pairwise correlation. The updated results for this study range from 28-85% (mean = 52%) for mosquitoes and 33-88% (mean = 53%) for human disease cases. This indicates that the model parameterized from lab data can account for more than half of the variation in dynamics that we observed in the field. We believe that the updated data and figure provide more substantial evidence of model robustness.

Reviewer # 2 Comment # 2: How was 'catchability' dealt with, in particular for model validation? Vector/larvae surveillance captures only a very small portion of the true population, both in terms of numbers and also the life history stage (e.g. host-seeking). Also, how is the information from sampling the breeding sites included? It is also not clear if the model developed and data used to validate it are for Ecuador and Kenya in isolation or combined or for single arbovirus or all three combined. Throughout the results, discussion and methods there are multiple instances where could be both. I think most results are averages across all sites but some parameters seem to common. Different diseases will behave differently within region, let alone between the two regions. This should all be thoroughly clarified throughout the manuscript.

Response: We used the same trapping methods for each life history stage across all study sites with many replicate traps within sites. Therefore, 'catchability' should not be an issue since the methods used were the same at each location and survey time. However, it is true that vector surveillance

captures a very small portion of the true population. This is a very important point if we want to predict the absolute magnitude of mosquitoes at any point in time. For this reason, we only looked at relative vector dynamics for comparison with model predictions (i.e., pairwise correlation). We have added text to clarify this in Methods section on vectors. We also now clarify how we used the data to validate the model. We now use the term “pairwise correlation” throughout the text, to indicate that we compared a time series of model predictions with a time series of observations for each site and response variable (vector abundance or human disease cases).

Other comments:

Reviewer # 2 Comment # 3: In the introduction the authors mention seemingly contradictory results (e.g. L73) in other climate-driven papers but it is not clear what type of contradictions they are talking about. Please provide examples and how this paper will add to this. Examples in L80 might not be so useful without specific temperature ranges and some context. An explicit statement about what exactly is novel about your framework would be very useful. There are multiple publications showing non-linear relationships between mosquito abundance and temperature and temperature optimums, including for *Aedes* sp. (e.g. Afrane et al 2012, Sharmin et al 2018, Trewin et al 2019, Mordecai et al 2019 etc) that should be added and already support the hypothesis in L81-84. What did these studies lack that your provide?

Response: We have revised the sentence referenced in this comment, and throughout the text, better clarify the novelty of this study and how this work builds on other research (as suggested).

Reviewer # 2 Comment # 4: The only sensitivity analysis done seems to have been on the initial conditions. All the remaining parameters seem to be static means. We know that small variations in some of the life history parameters could lead to changes in the vector population and incidence dynamics. Also, importantly, was there any sensitivity analysis done on the climate-dependent parameters? Could the model be overparameterized? Not only rainfall and temperature are often highly correlated, there could be confounding factors between most of the temperature dependent traits defined (L494-507).

Response: This is a good point. We added a sensitivity analysis using 50 random samples from the posterior distributions of all the temperature dependent traits. We have added figures of the results to the supplemental materials (Figs. S9-10) to show how the time series of predictions differ for each model run. Overall, we find that the mosquito population trajectories are relatively insensitive to small variations in life history parameters. The trajectories of predicted cases similarly show the same dynamics regardless of small variations in the life history parameters, although these variations do affect the magnitude of cases.

Although there are numerous temperature dependent traits included in the model that could be correlated with each other and with rainfall, we do not believe overparameterization is an issue for a mechanistic model. The reason we include so many temperature dependent traits is because we know each of them are important for transmission dynamics, and because they are independently supported by laboratory experimental data (as summarized in Mordecai et al. 2017). Further, overparameterization is primarily a concern for overfitting a model to data. Since we are evaluating model fit with completely independent data (field observations), there is no concern of overfitting the model to data and therefore having overconfidence in our results.

Reviewer # 2 Comment # 5: Abstract: Objective not clear in the abstract.

Response: Thank you for this comment, we have revised the Abstract to clarify the objective.

Reviewer # 2 Comment # 6: L70-73: Provide references

Response: We have added references to this sentence.

Reviewer # 2 Comment # 7: L116-118: Two out of three objectives are methodological. The introduction is lacking information on other similar modelling approaches.

Response: We have revised the objective sentence to focus on the key questions this study addresses, and we added a sentence about previous similar modeling approaches to the Introduction.

Reviewer # 2 Comment # 8: Fig. 2. Schematic is good but perhaps missing the key part of the paper which is where climate is impacting the mosquito traits and which disease transmission parameters it influences. Can the schematic be expanded to include this?

Response: Thank you for this suggestion. We have revised Figure 2 to show the traits that vary with temperature, humidity, and rainfall.

Reviewer # 2 Comment # 9: L261: But temperature in Equador doesn't seem to vary that much.

Response: This is a good point, we have removed the comment about temperature variability in this sentence.

Reviewer # 2 Comment # 10: Fig. 5. Please expand explanation of figure.

Response: We have removed Fig. 5 from the paper because we now use a different statistical method to compare model predictions and observations based on reviewer comments.

Reviewer # 2 Comment # 11: Fig. 6. The arrows indicate the direction of what?

Response: The arrows indicate whether the direction of the relationship is positive (north facing arrow) or negative (south facing arrow). We have added a legend for clarification.

Reviewer # 2 Comment # 12: L291. Provide levels as reader won't have read the methods yet.

Response: Thank you for this suggestion, however, we removed the intervention analysis from the paper based on multiple comments, therefore, this edit is no longer necessary.

Reviewer # 2 Comment # 13: Fig. 7. Hardly any impact on cases with a 90% reduction in population size is surprising. Could these results be because we don't know the relationship between mosquito abundance and arbovirus cases? If so, what is the implication for the model developed here?

Response: We have removed Fig. 7 from the paper because we no longer present an intervention analysis based on multiple reviewer comments. One explanation for why we saw little impact on cases with a 90% reduction in population could certainly be associated with our limited understanding of the direct relationship between mosquito abundance and arbovirus cases.

Reviewer # 2 Comment # 14: L329. Not sure I follow this argument as you only use on metric too, no?

Response: This is a good point. We have removed this sentence based on this comment and another reviewer comment suggesting this section is too redundant with the Introduction.

Reviewer # 2 Comment # 15: L335-338. What in your results specifically highlights this?

Response: This comment refers to idea that climate and climate lags may differentially impact disease dynamics in different settings. This statement is now better supported by a new figure (Fig. 6) and paragraph in the Discussion.

Reviewer # 2 Comment # 16: L419. Why before interpolations? How do you handle the NA's?

Response: To fill in missing rainfall data, we regressed 14-day cumulative rainfall from rain gauge measurements with modeled rainfall. We calculated the 14-day rainfall before interpolation because daily rainfall values are less reliable than two-week rainfall accumulation. We excluded any days with missing data (NAs) from the 14 days prior to the observation from the linear regression, and then used the linear regression to interpolate 14-day accumulated rainfall based on modeled rainfall. We have added text to the Methods climate data section to explain the motivation for making this calculation prior to interpolation and how we handled NAs.

Reviewer # 2 Comment # 17: Data. It would be good to see a Figure with the data.

Response: Thank you for this suggestion. We have replaced the scatterplots of predictions versus observations for all sites and time periods with site-specific time series plots of model predictions and field observations for mosquitoes and human disease cases (Fig. 3).

Reviewer # 2 Comment # 18: Section SEI-SEIR. It would be helpful to define the equations in order, instead of the bottomup, I,e, equations are mosquito to human, text written human to mosquito.

Response: We now present the results for mosquitoes first, followed by human disease cases, to reflect the order of the SEI-SEIR model equations.

Reviewer # 2 Comment # 19: L509. Is K for egg to adult stages? all combined? Do you think the simplification of the mosquito life cycle (e.g. no larval stage) has implications for the results?

Response: Carrying capacity, K, is for the adult mosquito population. We have added text to clarify that K refers to the entire adult mosquito population in Fig. 2 and in the Methods. We include a parameter to account for egg to adult development in the model that accounts for the time delay between egg laying and adult emergence and allow for associated mosquito mortality. We simplified the mosquito life cycle because we know temperature affects mosquito traits in each life history stage, but we do not have laboratory data available to parameterize those traits, thus we chose to use an integrated temperature dependent development rate. Although a model that explicitly uses each life cycle stage may be more accurate, we do not believe a model that includes those life cycle stages would lead to substantially different results from the model we present.

Reviewer # 2 Comment # 20: L512. I'm confused, surely the 29 degrees are context dependent. Also, if this is one of the parameters, how can it be a finding too?

Response: L512 refers to the optimal temperature incorporated in the carrying capacity equation, which is used to determine the maximum number of mosquitoes in the system at any one time. This is not a finding, but an input value. We use 29 degrees Celsius based on the physiology of *Aedes aegypti*, as in previous work (Mordecai et al. 2017 PLOS NTD; Huber et al. 2018 PLOS NTD). We do not expect this value to be context-dependent because it is related to physiological constraints on mosquitoes, which are considered relatively stable across populations and regions. We have expanded this sentence to clarify that these input values are based on physiological optimal conditions measured in the laboratory.

Reviewer # 2 Comment # 21: L588. What is a direction trend?

Response: We have removed this sentence from the manuscript as it referred to the previous statistical analysis for comparing model predictions with field observations, and is therefore no longer relevant.

Reviewer # 2 Comment # 22: L614. Is there data to look at rainfall too? So much work is done on temperature but it seems that rainfall is a key part of the system.

Response: This is a great suggestion, thank you. We have created a conceptual figure to complement the temperature work that we presented in the original manuscript based on this study and prior studies.

Reviewer # 2 Comment # 23: Table S2: This should be presented as percentages as in main text Table 1

Response: Thank you for this suggestion. We no longer present Table S2 because we have changed the analysis for comparing predictions and observations, however, we have ensured that the results are reported consistently throughout the manuscript.

Reviewer #3 (Remarks to the Author):

The manuscript presents an interesting analysis of the mechanistic effect of climate on transmission of arboviral infections. By mechanistically linking temperature-mosquito traits relationships to a dynamic compartmental model of transmission of arboviruses, the authors investigated how well climatic factors such as temperature, humidity and precipitation drives observed patterns of dengue, zika and chikungunya cases in two settings of Kenya and Ecuador. I find the study novel and of interest to the scientific community. I have a few comments which might help strengthening the aims and conclusions of the paper:

Reviewer # 3 Comment # 1: The introduction to the study stresses on the

idiosyncratic nature of climate effects of disease transmission, e.g. lines 69-84. However, I find that the results, apart from the rainfall-mosquito relationship (Table 2), do not emphasize much this aspect, specifically the fact that the effect of climate on arbovirus transmission in Kenya and Ecuador is context-dependent. I would find it interesting to see more results on this. For instance, Figure 4 shows model predictions and observed number of cases for two settings in Ecuador. I would like to see how model predictions compare to observed number of cases in Kenya as well. Furthermore, while a measure of overall fit by settings/sites is reported in the supplementary material, I think it is important for the purpose of applying the results to other settings, to also show how the model predicts observed data at different times throughout the epidemics. I would also like to know a bit more on the climatic conditions leading to the model under predicting and over predicting in different sites in Ecuador and Kenya. It seems that the CART analysis was conducted on the overall dataset. A CART analysis conducted for the different settings would be quite informative on the aspects which make the effect of climate unique across settings. Generally I think it would be useful, and it would help the paper to align to its initial claims regarding the scope of the research, if the authors could highlight a bit more in the results the differential effects of climate across sites and expand on the causes of these differences in the discussion.

Response: Thank you for this comment. We have revised the paper in several ways to better focus on the idiosyncratic, yet mechanistically predictable, nature of climate effects and other factors on disease transmission. We created a new figure to show how model predictions and field observations align for each site through time (Fig. 3). We also revised the figure that conceptually showed the idiosyncratic nature of temperature on transmission to include a similar conceptual figure for rainfall (Fig. 6). Further, throughout the main text we have focused more on how several factors, including climate, impact our ability to predict disease dynamics. We like the idea of a stratified CART analysis by site, however, in response to several reviewer comments, we have changed our method for comparing model predictions with observations and that new method does not produce outputs that are appropriate to use in a CART analysis. Instead, we have added a figure that emphasizes patterns in the predictability of vector and disease dynamics across a variety of climate, social, and ecological factors across sites (Fig. 4), which we believe helps highlight the differential effects across sites.

Reviewer # 3 Comment # 2: I am curious to why the model does not seem to predict dengue cases as well as it seem to do with Zika and Chikungunya. From Figure 4 (although it is not clear as observed cases are aggregated across diseases) it looks like the model over predicts dengue cases at quite a few times. Can the author provide some justification to why they think this is the case? Also, it would be useful to see a time-dependent figure of the comparison between observations and model predictions shown for the different viruses.

Response: Although the idea that the model may predict some diseases better than others is interesting, we do not have enough data to make an inference about this point. We now show a time-dependent figure of the comparison between observations and model predictions across all sites to help readers better evaluate the relationship between predictions and observations over time (Fig 3). In addition, we now present aggregated arbovirus incidence to avoid confusion about predictability across diseases since (1) the model is parameterized to predict any *Aedes*-transmitted disease and (2) we only have observations of multiple arboviruses in two of the eight study sites. We do, however, think there is potential for the model to better predict certain arboviruses over others, especially if we were to conduct experiments to better parameterize the virus development rate for each disease and consider potential effects to mosquito behavior when infected with each disease. We have therefore added text about this idea to the discussion.

Reviewer # 3 Comment # 3: I am a bit unsure regarding the practical applicability of the intervention simulations. To simulate the effect of interventions the authors varied three mosquito traits (mosquito biting rate, carrying capacity and mosquito abundance) by different amounts and recalculated the total number of cases by running forward the compartmental model. Intervention assessment can also be done by varying epidemiologically variables such as the basic reproduction number of the disease, R_0 , which can be reasonably estimated from national disease surveillance system. In this case, the authors chose to vary mosquito traits, and this in reality would require information on these traits, which might not be available. Some discussion around this issue would be useful. Also it would be fairer to clarify that the results of this intervention modelling exercise (specifically the reduction in burden) are purely based on lab data and simulation and they do not include any data collected in the field (e.g. disease case number). They were not tested or validated. This obviously limits the degree to which they can be trusted or extrapolated to other settings. Providing a bit more context, for instance on why certain effect sizes (10,50,90%) were simulated, how they can be measured (need to have information on those traits in the field) and how they be implemented in practice would help clarifying the relevance of this analysis and strengthen the applicability of its results.

Response: Thank you for this comment. We agree with the concerns of all the reviewers that the practicality of the intervention simulations is limited given the lack of validation data and we have therefore decided to remove this analysis from the paper.

Some minor comments:

Reviewer # 3 Comment # 4: Line 57. It is a bit vague. What are the contradictory results and how this paper helps resolving them?

Response: We have revised this sentence to include specific details about our model results and how they show that climate-driven mosquito traits alone can produce a wide range of transmission dynamics that occur across ecologically distinct settings.

Reviewer # 3 Comment # 5: Line 59 “tested”. I would tone this down a bit – the intervention results come from untested effect sizes and they might be difficult to use in reality due to general lack of field data on mosquito traits in the field.

Response: We agree with this comment and other comments regarding the intervention analysis, so we have removed this analysis from the paper.

Reviewer # 3 Comment # 6: Line 91. Another control strategy for Aedes-borne viruses is *Wolbachia*. Maybe worth mentioning it.

Response: This is a good point. We now mention *Wolbachia* in this sentence.

Reviewer # 3 Comment # 7: Line 120. What is the original study (reference) of the model used in this paper?

Response: We have added a reference to the original study (Huber et al. 2018) in line 120.

Reviewer # 3 Comment # 8: Lines 124-138. These distinctions between study settings are interesting. It would be interesting to see in the results a stronger emphasis on the distinct effects of climate on disease transmission across settings. Otherwise at the moment, this session does not seem to have a strong reason to be here.

Response: Thank you for this comment. We now add a table, figure, and text in the Results and Discussion section to emphasize distinctions among study settings and what factors may influence our ability to predict disease dynamics from a climate-driven model.

Reviewer # 3 Comment # 9: Line 151 “Site-months”. Are predictions made by months? I find the term unclear.

Response: We no longer use the term “Site-months” in the manuscript both for clarity and because we changed the method we use to compare model predictions with observations.

Reviewer # 3 Comment # 10: Figure 2. Perhaps consider colouring only the boxes rather than whole areas – to highlight the different elements of the model (not just the host and the vector)

Response: We have removed the colors from Figure 2.

Reviewer # 3 Comment # 11: Figure 3. The title of panel b should be “cases”? It would be interesting to see this correlation performed separately for Ecuador and Kenya, if possible.

Response: We have replaced Figure 3 with time series plots for predictions and observations of mosquito and human disease dynamics for each site and made sure to use appropriate titles.

Reviewer # 3 Comment # 12: Figure 4. What is the fit between model predictions and the observed temporal trends of cases in Kenya?

Response: We now present the model fits for each site in Figure 3 and Table 2. The pairwise correlation between model predictions and arboviral cases in Kenya varies between 36 and 59%.

Reviewer # 3 Comment # 13: Figure 4. Also, can model predictions can be disaggregated by disease? It is difficult to compare them with data shown by disease.

Response: The model does not make different predictions for dengue, chikungunya, and Zika, but rather, predicts the risk of any arboviral infection. To avoid confusion, we now present aggregated disease observations.

Reviewer # 3 Comment # 14: Line 290. “Predicting” might be more appropriate here than “evaluating”.

Response: We have removed this section from the paper because we no longer present the intervention analysis based on multiple reviewer comments.

Reviewer # 3 Comment # 15: Line 323-335. This part sounds a bit like a repetition of the introduction. The authors might consider expanding here on their results and the novelty of the study.

Response: This is a good point. We now focus the Discussion more on the results and novelty of the study.

Reviewer # 3 Comment # 16: Line 340-342. I would highlight these important implications in the introduction.

Response: Thank you for this suggestion, we have added text to the Introduction to highlight the usefulness of mechanistic models to address critically important research priorities.

Reviewer # 3 Comment # 17: Line 566. Why were these effect sizes chosen? Is there any indication that any of these values is a realistic target to aim for?

Response: We used a random sample function to select the initial population proportions for each compartment. We selected these proportions (“effect sizes”) from the range of initial condition proportions we tested in a sensitivity analysis. We have added text to clarify this point.

REVIEWER COMMENTS

Reviewer #1 (Remarks to the Author):

The authors have substantially revised and improved the manuscript. There are however some major conceptual issues and aspects of the results that need to be addressed and clarified.

Major comments:

1) The work is presented in the context of phenomenological vs. mechanistic models to motivate the work and explain its scope and relevance. This is an insufficient framework, which may confuse readers who do not themselves work with models. For this work, what seems most relevant is how the model is parameterized and not just that it is mechanistic. In particular, there are two general approaches one can distinguish to define what is unique about this work. That is, given a set of process-based or mechanistic equations, the authors use parameters obtained independently from the transmission system and data they wish to predict/explain. These parameters come primarily from the laboratory but also from the literature and therefore, other locations. In contrast, most studies applying process-based models to specific data would infer some parameters by fitting the model to these data. The degree to which a subset of parameters is fixed to "known" or laboratory-based values would vary. In essence, there would be some calibration of the model to the data. Regardless of the specific statistical method of inference, the approach would allow for under-reporting (with some assumption for measurement error), and besides the parameters themselves one would have to include inference of initial conditions. The resulting models would be called mechanistic and can be used for prediction. The distinction I make here is important because as I said in my previous review what the authors try to do, by using no calibration at all, is not what would generally be done. As I said before, there are many good reasons for this, and I will not enter here on the philosophy of modeling, and the degree to which one can expect parameters obtained at the level of individuals in the lab to apply exactly as such to populations in the field at very different spatial scales. Since any model structure is at best a crude representation of reality, parameter values often need to take values that are not the basic biological ones (from direct measurement, say in the lab or even the field) to account for discrepancies with population data. Nevertheless, it can be interesting to ask and evaluate what a completely "bottom up" model like the one parameterized here can achieve, since one would not expect to work too well. Still, it cannot be done in the simplistic dichotomy of mechanistic vs. phenomenological. The parameterization of mechanistic models is the important consideration, which needs a much more clear presentation. If what the authors attempt works, the value might be primarily for (a) future scenarios, where calibration/fitting in the present may be insufficient; (b) for locations where data is unavailable. For explanation and seasonal prediction, including early warning systems, one would expect models fitted to data for the specific locations to be the best approach. Model selection criteria based on inference methods further allow the rigorous comparison of models and associated hypotheses. The authors are probably aware of most of what I say here but it is not apparent from the writing and the way conclusions are drawn.

2) The above point is strongly connected to another important issue. How to evaluate the models? The authors have improved on this by replacing the results based on z-scores with the temporal correlations. This is still a limited way to go. The correlation would concern temporal variability in a general way. It does not address the overall levels of disease or vectors, that is, the change in the mean cases, or mean vectors, across the sites. This would be of interest because it asks not whether simulations covary linearly in time with data, but whether they can account for the between sites differences. The different locations span different altitudes (and therefore temperatures) and different degrees of urbanization. There is no explicit treatment of whether these two important gradients for these vector-borne diseases of urban environments generate differences in overall incidence and vector abundance that the models can predict. (One may argue that since there is no explicit consideration of measurement errors/reporting rates, an evaluation of this on the basis of absolute values is not warranted. Then, one can ask how the trends in mean

levels across locations, with the average being taken over time, compare between models and data. Are there trends with altitude and urbanization that the model captures?). In brief, given how unusual the approach of not fitting any parameter is, it becomes interesting and important to think about what aspects of the data and of the differences in dynamics, the models are able to capture. The correlations focus on associations/co-variation over time in one site. What else can be evaluated? Trends across sites is clearly one worth considering given the strong altitude changes. Other aspects that are included but one may want to differentiate, might be seasonality, timing of large outbreaks.

3) Related to (1) and (2), the way the initial conditions are set is extremely important. The way this is done here is quite idiosyncratic. This is where inference/fitting would be most useful, together with obtaining values for the parameters that influence the carrying capacity (the rate constant c , the maximum number of vectors per human). The most reasonable approach would be to estimate these once other parameters more easily obtained from the lab, are set. A generic parameterization of the effect of rainfall on carrying capacity is very likely to vary strongly across space with myriad factors there is no way to control for. Since the approach is not here one of inference, greater care, analysis and discussion of the way these parameters are set is warranted.

4) The baseline comparison to a random pattern is a very low bar. It would be valuable to include as a reference something that is less trivial to improve upon. An example would be a seasonal autoregressive model fitted to the data, but the authors may have other ways to do this.

5) Is the general model of how rainfall enters in the dynamics formulated for mean/total rainfall? The way water storage plays in this pattern will depend strongly also on seasonality and other aspects of variability, as well as on how predictable is the supply of water to people.

Minor comments:

5) We are told that the regions are hyperendemic but then that transmission is low in Kenya. This appears contradictory.

6) Urbanization values: is there a way to interpret this scale, beyond relative comparisons?

7) The lower "Sensitivity/response" to temperature close to the maxima, follows from the shape of R_0 with this climate factor, which in turn, defines the slope. I do not think this deserves so much explanation with no mention of this simple aspect.

8) The conclusion in line 308 should be rewritten. It is very weak and confusing as stated: "Our results highlight that we should not expect the same climate conditions and lags to be important in all settings, but that their combined, nonlinear effects can generally predict disease dynamics across different ecological and socio-economic settings". In fact, the climate conditions differ across locations. If these were the same and everything else were the same, we could expect similar dynamics. It is not about combined nonlinear effects per se, but that these are operating in different contexts (socio-economic, human movement, human behavior affecting water storage etc). This is why in the end most models of complex biological systems require some calibration. I could not clearly understand what was meant by this conclusion. Some rewriting will make the text more convincing.

Reviewer #2 (Remarks to the Author):

The revised version of the manuscript is much clearer and addresses some of the concerns from

the previous review.

However, the main concern over the whether the model can explain the observed data remains. I agree with the authors that the revised approach to assess this, i.e. correlation test between observed and predicted dynamics; is now clearer and more robust, and that they find a significant correlation in most sites for both vector and disease dynamics is encouraging. What concerns me, is the magnitude of these correlations which average just under 0.5 (though going as low as 0.28 and as high as 0.85). Doesn't this mean that the majority of the variability is not explained and so predictions from this model become uninformative? This underfit seems to be a problem of magnitude but more importantly also of pattern, where peaks of abundance/incidence are missed or offset. Given that the premise of the manuscript is predictability, not being able to predict these (or on average half the dynamics) correctly seems like a critical problem given the premise of the manuscript. For describing dynamics adding a caveat that only partly is explained seems understandable but for prediction this becomes more complicated.

The removal of the intervention section released space to further explore the drivers of the dynamics. I believe this change strengthens the manuscript considerably. However, I couldn't understand the methodology here. How are these factors included in the model and is their relative importance assessed? I found this confusing for two main reasons: i) I thought model was fit and assessed for each site individually, but these factors seem to operate on all sites combined. ii) model predictability seems to be assessed by correlation tests but the values of the test for these factors are not provided; i.e. insecticide usage is removed from the model how much does the correlation value and p-value change? Or was this assessed using a different metric? Also, with such low sample size from sites with different dynamics can local drivers and general predictors be distinguished?

Other comments:

L74: what mechanistic effect?

L88-89: Second part of the sentence (after comma) is repetition of the first. Could delete to improve clarity.

L121: Mechanistic models of what? mosquito life cycle, disease dynamics both? for Aedes or other vectors?

Table I: Is bednet usage relevant for Aedes?

L173-176: Better than 0% correlation seems like a low bar given that predictability is the key aim of the manuscript.

L252-254: Not sure I understand how figure 5 shows that predictability decreases near optimal temperature? How does the effect size and direction suggest predictability? Should this be based on the correlation coefficient with the raw data?

L255-256: the introduction suggests this is unknown.

L261: How is this conceptual model different from what has already been defined in the intro with references?

Discussion:

- The implication of assessing all diseases combined should be discussed. How useful is a model that combines all arboviral pathogens?

- If this is a trait-based model there needs to be a discussion of the parameterization of these traits. Are they biologically meaningful? How much impact do these assumptions have for the vector and disease etc.

L314: the accuracy wasn't really assessed, was it?

L316-317: and a low sample size.

L329-334: Could also be because of the low predictability overall? The model is not able to predict at all temperatures for example.

REVIEWER COMMENTS

Reviewer #1 (Remarks to the Author):

The authors have substantially revised and improved the manuscript. There are however some major conceptual issues and aspects of the results that need to be addressed and clarified.

Major comments:

Reviewer #1 Comment #1: The work is presented in the context of phenomenological vs. mechanistic models to motivate the work and explain its scope and relevance. This is an insufficient framework, which may confuse readers who do not themselves work with models. For this work, what seems most relevant is how the model is parameterized and not just that it is mechanistic. In particular, there are two general approaches one can distinguish to define what is unique about this work. That is, given a set of process-based or mechanistic equations, the authors use parameters obtained independently from the transmission system and data they wish to predict/explain. These parameters come primarily from the laboratory but also from the literature and therefore, other locations. In contrast, most studies applying process-based models to specific data would infer some parameters by fitting the model to these data. The degree to which a subset of parameters is fixed to “known” or laboratory-based values would vary. In essence, there would be some calibration of the model to the data. Regardless of the specific statistical method of inference, the approach would allow for under-reporting (with some assumption for measurement error), and besides the parameters themselves one would have to include inference of initial conditions. The resulting models would be called mechanistic and can be used for prediction. The distinction I make here is important because as I said in my previous review what the authors try to do, by using no calibration at all, is not what would generally be done. As I said before, there are many good reasons for this, and I will not enter here on the philosophy of modeling, and the degree to which one can expect parameters obtained at the level of individuals in the lab to apply exactly as such to populations in the field at very different spatial scales. Since any model structure is at best a crude representation of reality, parameter values often need to take values that are not the basic biological ones (from direct measurement, say in the lab or even the field) to account for discrepancies with population data. Nevertheless, it can be interesting to ask and evaluate what a completely “bottom up” model like the one parameterized here can achieve, since one would not expect to work too well. Still, it cannot be done in the simplistic dichotomy of mechanistic vs. phenomenological. The parameterization of mechanistic models is the important consideration, which needs a much more clear presentation. If what the authors attempt works, the value might be primarily for (a) future scenarios, where calibration/fitting in the present may be insufficient; (b) for locations where data is unavailable. For explanation and seasonal prediction, including early warning systems, one would expect models

fitted to data for the specific locations to be the best approach. Model selection criteria based on inference methods further allow the rigorous comparison of models and associated hypotheses. The authors are probably aware of most of what I say here but it is not apparent from the writing and the way conclusions are drawn.

Response: The Reviewer brings up an excellent point and we have revised the manuscript considerably to address this concern with updated text and a new analysis and figure (new Fig. 3). We appreciate the Reviewer further explaining this concern raised in the initial set of reviews, which helped us to more fully respond to it. In the revised manuscript, we use the model results to ask how well a model with limited calibration (we clarify that we did use some model selection with rainfall functions and time lags) captures key epidemic characteristics associated with the number, timing, duration, and magnitude of outbreaks. We hypothesized that the model should capture the timing and duration of outbreaks but not the magnitude of outbreaks based on a previous simulation study. We are very excited that our new results support those hypotheses and also indicate that the model captures as additional epidemic characteristic - the number outbreaks - within sites. We also believe the updated framing better reflects our thought process that initially motivated the study, as we did question whether this type of model could reproduce any of the dynamics we see in the field, given the many different factors that are important for transmission.

Reviewer #1 Comment #2: The above point is strongly connected to another important issue. How to evaluate the models? The authors have improved on this by replacing the results based on z-scores with the temporal correlations. This is still a limited way to go. The correlation would concern temporal variability in a general way. It does not address the overall levels of disease or vectors, that is, the change in the mean cases, or mean vectors, across the sites. This would be of interest because it asks not whether simulations covary linearly in time with data, but whether they can account for the between sites differences. The different locations span different altitudes (and therefore temperatures) and different degrees of urbanization. There is no explicit treatment of whether these two important gradients for these vector-borne diseases of urban environments generate differences in overall incidence and vector abundance that the models can predict. (One may argue that since there is no explicit consideration of measurement errors/reporting rates, an evaluation of this on the basis of absolute values is not warranted. Then, one can ask how the trends in mean levels across locations, with the average being taken over time, compare between models and data. Are there trends with altitude and urbanization that the model captures?). In brief, given how unusual the approach of not fitting any parameter is, it becomes interesting and important to think about what aspects of the data and of the differences in dynamics, the models are able to capture. The correlations focus on associations/co-variation over time in one site. What else can be evaluated? Trends across sites is clearly one worth considering given the

strong altitude changes. Other aspects that are included but one may want to differentiate, might be seasonality, timing of large outbreaks.

Response: We appreciate the Reviewer's excellent suggestions. We now present the new analyses (described above) showing that the model captures the timing, number, and duration of outbreaks but not differences in magnitude across sites. In comparing magnitude, we focus on patterns across time and space and not on the absolute magnitude of the predictions and observations, thereby allowing for under-reporting rather than explicitly estimating a reporting rate or function. In addition, we now provide statistics for the comparison between pairwise correlation values with a variety of socio-economic factors listed in Table 1 to account for differences across sites. We found significant relationships with demography, housing construction, and temperature (updated Fig. 6).

Reviewer #1 Comment #3: Related to (1) and (2), the way the initial conditions are set is extremely important. The way this is done here is quite idiosyncratic. This is where inference/fitting would be most useful, together with obtaining values for the parameters that influence the carrying capacity (the rate constant c , the maximum number of vectors per human). The most reasonable approach would be to estimate these once other parameters more easily obtained from the lab, are set. A generic parameterization of the effect of rainfall on carrying capacity is very likely to vary strongly across space with myriad factors there is no way to control for. Since the approach is not here one of inference, greater care, analysis and discussion of the way these parameters are set is warranted.

Response: This is a great point, which we spent substantial time thinking about as the research progressed, but perhaps did not clearly explain in the manuscript. For the initial conditions, we now clarify that we conducted sensitivity analyses to determine how different initial conditions, across the range of possible conditions, affected the system, because for emerging diseases the system can be highly sensitive to initial conditions and lead to chaotic dynamics. We found that the model converged on the same dynamics regardless of initial conditions after a three month burn-in period (Fig. S10) and therefore the dynamics in this study are stable. In addition, we clarify that we set the ratio of vectors to humans based on previous research (line 615). We also revised the text to explain that since c is multiplied by a scaling factor, it is adjusted within this model (lines 581-584).

Reviewer #1 Comment #4: The baseline comparison to a random pattern is a very low bar. It would be valuable to include as a reference something that is less trivial to improve upon. An example would be a seasonal autoregressive model fitted to the data, but the authors may have other ways to do this.

Response: Motivated by the Reviewer's very helpful comments, we now show that the model is capturing several real epidemic patterns in the data. In the

previous manuscript, we really were not considering random as a null model and we have removed the sentence that suggests this line of thinking (e.g., where we say a random model would result in $r = 0$). In addition, we have added text to explain that we would never expect a perfect fit to data, even if we were able to perfectly capture the true climate dynamics in the model (since so many other factors are important for transmission). Since we do not know what portion of disease dynamics are truly driven by climate, we do not think there is a truly good null model to use as a comparison in this study. We now clarify this thinking in the main text (lines 359-364).

Reviewer #1 Comment #5: Is the general model of how rainfall enters in the dynamics formulated for mean/total rainfall ? The way water storage plays in this pattern will depend strongly also on seasonality and other aspects of variability, as well as on how predictable is the supply of water to people.

Response: We now clarify that we formulated rainfall as the total accumulated rainfall over a 14-day moving window, which is a short enough time window to capture variability and seasonality in rainfall patterns (lines 464-466). The Reviewer brings up an important point in this comment about the many ways in which rainfall can indirectly influence transmission. We tested the inverse rainfall function in this study as a proxy for human water storage practices in locations without access to piped water or with an unreliable water supply. We have added sentences to the main text to expand upon the motivation for the 14-day time window and how the inverse rainfall function reflects the effects of human behavior on the relationship between rainfall and transmission.

Minor comments:

Reviewer #1 Comment #6 We are told that the regions are hyperendemic but then that transmission is low in Kenya. This appears contradictory.

Response: We have removed the term hyperendemic from this sentence and revised it to indicate that all four dengue serotypes are co-circulating in both countries.

Reviewer #1 Comment #7: Urbanization values: is there a way to interpret this scale , beyond relative comparisons?

Response: We agree that the urbanization index, as measured in units of light per unit area ($nW \cdot cm^2 \cdot sr^{-1}$), was not very intuitive. This metric was meant to be a proxy for the built environment. Since the correlation between model predictions and model observations did not vary in any systematic way with this metric, and a combination of other metrics (e.g., dominant land cover type, population size, housing materials, etc.) are also indicative of urbanization, we have removed this variable from Table 1.

Reviewer #1 Comment #8: The lower “Sensitivity/response” to temperature close to the maxima, follows from the shape of R_0 with this climate factor, which in turn, defines the slope. I do not think this deserves so much explanation with no mention of this simple aspect.

Response: We have added text to indicate that the response to temperature follows the shape and slope in the R_0 curve as recommended by the Reviewer (lines 311-315), and we consolidated other sentences that address this idea.

Reviewer #1 Comment #9: The conclusion in line 308 should be rewritten. It is very weak and confusing as stated: “Our results highlight that we should not expect the same climate conditions and lags to be important in all settings, but that their combined, nonlinear effects can generally predict disease dynamics across different ecological and socio-economic settings”. In fact, the climate conditions differ across locations. If these were the same and everything else were the same, we could expect similar dynamics. It is not about combined nonlinear effects per se, but that these are operating in different contexts (socio-economic, human movement, human behavior affecting water storage etc). This is why in the end most models of complex biological systems require some calibration. I could not clearly understand what was meant by this conclusion. Some rewriting will make the text more convincing.

Response: We appreciate the Reviewer highlighting a sentence that was unclear in the discussion. We have removed this sentence from the text and instead expand and focus on the most exciting findings from the study, including the new results described above: (Lines 350-355) “The trait-based modeling approach captured several key epidemic characteristics and generated a range of disease dynamics along a spectrum of settings with low levels of transmission to seasonal outbreaks, helping to reconcile seemingly context dependent effects (i.e., opposite conclusions about the magnitude and direction of effects; Fig. 7) of climate on arboviral transmission dynamics from the literature [7–12,47].”

Reviewer #2 (Remarks to the Author):

The revised version of the manuscript is much clearer and addresses some of the concerns from the previous review.

Reviewer #2 Comment #1: However, the main concern over the whether the model can explain the observed data remains. I agree with the authors that the revised approach to assess this, i.e. correlation test between observed and predicted dynamics; is now clearer and more robust, and that they find a significant correlation in most sites for both vector and disease dynamics is encouraging. What concerns me, is the magnitude of these correlations which average just under 0.5 (though going as low as 0.28 and as high as 0.85). Doesn't this mean that the majority of the variability is not explained and so predictions from this model become uninformative? This underfit seems to be a problem of magnitude but more importantly also of pattern, where peaks of abundance/incidence are missed or offset. Given that the premise of the manuscript is predictability, not being able to predict these (or on average half the dynamics) correctly seems like a critical problem given the premise of the manuscript. For describing dynamics adding a caveat that only partly is explained seems understandable but for prediction this becomes more complicated.

Response: These are very good points that were also raised by the other Reviewer. In response to several comments, we have added a new analysis and figure to identify key epidemic characteristics captured by the model, which is parameterized with data from other studies. These new results indicate that the model is more adept at predicting the number, timing, and duration of outbreaks compared with predicting the magnitude of outbreaks. We now quantify the differences in peak timing of outbreaks (Fig. 3b) as well as the number of outbreaks that are missed (Lines 188-192). In addition, we present correlations within sites as a metric for the proportion of the true disease dynamics that we might expect to capture in new settings. This is quite useful because few studies use true out-of-sample validation. Since it is not feasible to study arbovirus dynamics in detail in every possible transmission setting, one of our goals was to examine how much information about epidemic dynamics can be captured from a general, climate-driven model like the one we present here. As we now clarify in the text, we would never expect a climate-driven model to perfectly reproduce dynamics in the field as many other factors influence transmission. As a result, correlations between 0.28 and 0.85 were actually higher than we originally expected, as the worst model would produce correlations values of -1. Therefore, an average correlation around 0.5 suggests that we can capture about half of all variation in disease dynamics with a climate-driven model.

Reviewer #2 Comment #2: The removal of the intervention section released space to further explore the drivers of the dynamics. I believe this change strengthens the manuscript considerably. However, I couldn't understand the methodology here. How are these factors included in the model and is their

relative importance assessed? I found this confusing for two main reasons: i) I thought model was fit and assessed for each site individually, but these factors seem to operate on all sites combined. ii) model predictability seems to be assessed by correlation tests but the values of the test for these factors are not provided; i.e. insecticide usage is removed from the model how much does the correlation value and p-value change? Or was this assessed using a different metric? Also, with such low sample size from sites with different dynamics can local drivers and general predictors be distinguished?

Response: We thank the Reviewer for highlighting a point of confusion. The way we assessed these relationships was through linear regressions comparing the site-level correlation between the SEI-SEIR model predictions and observations with site-level social and ecological factors (using a separate linear regression for each factor). We originally did not report statistics because of the low sample size (as mentioned by the Reviewer). However, since the original presentation led to confusion, we now add the results of the linear regressions to the manuscript to make it clearer that these results were from a separate analysis to the SEI-SEIR model.

Other comments:

Reviewer #2 Comment #3: L74: what mechanistic effect?

Response: We replaced “mechanistic effect” with “the effects of climate on population dynamics” for clarity.

Reviewer #2 Comment #4: L88-89: Second part of the sentence (after comma) is repetition of the first. Could delete to improve clarity.

Response: We removed the second part of the sentence as suggested.

Reviewer #2 Comment #5: L121: Mechanistic models of what? mosquito life cycle, disease dynamics both? for *Aedes* or other vectors?

Response: We edited this sentence to clarify that this study expands on previous work with models of the *Aedes* mosquito life cycle and disease dynamics.

Reviewer #2 Comment #6: Table I: Is bednet usage relevant for *Aedes*?

Response: We appreciate the Reviewer highlighting a potential point of confusion in the paper. Although there is some evidence that bed nets can be protective against *Aedes aegypti* for children when they sleep under them during naps, that effect would not apply to the general population. Therefore, we now clarify in the table caption that bed net use is a proxy for willingness to adopt intervention strategies rather than a specific adaptive response to *Aedes aegypti*, which bite during the day.

Reviewer #2 Comment #7: L173-176: Better than 0% correlation seems like a low bar given that predictability is the key aim of the manuscript.

Response: We agree and have removed this specific sentence from the study. Based on this comment and comments from the other Reviewer, we have reframed the expectation and results from the model more clearly.

Reviewer #2 Comment #8: L252-254: Not sure I understand how figure 5 shows that predictability decreases near optimal temperature? How does the effect size and direction suggest predictability? Should this be based on the correlation coefficient with the raw data?

Response: We have changed the placement of the parentheses in this sentence to better articulate that the reference to figure 5 (now Fig. 7) is in support of the 29 degree Celsius temperature optimum predicted from prior work, whereas the statement about predictability is based on results presented in this study (lines 311-316).

Reviewer #2 Comment #9: L255-256: the introduction suggests this is unknown.

Response: We have reworded this sentence for clarity. Our revised sentence indicates that studies in locations with temperatures around 29 degrees Celsius have found no or minimal effects of temperature on transmission.

Reviewer #2 Comment #10: L261: How is this conceptual model different from what has already been defined in the intro with references?

Response: Prior work that we reference in the Introduction provides evidence of multiple functional relationships for rainfall in different settings. Our results support multiple functional relationships even within the same settings, suggesting that these relationships may fall along a continuum rather than occurring as distinct relationships in specific settings. This conceptual model visualizes this relationship along a continuum. We have added a sentence to the manuscript to explain how this conceptual model adds to the field (lines 291-295).

Discussion:

Reviewer #2 Comment #11: The implication of assessing all diseases combined should be discussed. How useful is a model that combines all arboviral pathogens?

Response: We have added text to the Discussion regarding the limitations of assessing all diseases combined and how the virus-associated traits included in this model, namely parasite development rate and extrinsic incubation period, may differ among arboviral pathogens (lines 434-438).

Reviewer #2 Comment #12: If this is a trait-based based model there needs to be a discussion of the parameterization of these traits. Are they biologically meaningful? How much impact do these assumptions have for the vector and disease etc.

Response: Thank you for this suggestion, which complements many of the comments from the other Reviewer. We have edited the text throughout the manuscript to clarify that the traits in this model are parameterized from prior studies in the field and in laboratory experiments. We now focus some of the paper on answering the question of whether we should expect a model parameterized with data from other transmission settings to provide informative results. In addition, we have added text to explain that although laboratory experiments do not reflect real world conditions, the physiological responses measured are biologically meaningful (lines 547-548).

Reviewer #2 Comment #13: L314: the accuracy wasn't really assessed, was it?

Response: We have removed the term accuracy from this sentence. We now clarify that we are discussing the correlation between model predictions and observations.

Reviewer #2 Comment #14: L316-317: and a low sample size.

Response: We have edited this sentence to remind the reader that the correlation is for each site with a sample size of eight.

Reviewer #2 Comment #15: L329-334: Could also be because of the low predictability overall? The model is not able to predict at all temperatures for example.

Response: We now provide statistical results for this relationship and show that, despite a low sample size, the comparison between correlations and temperature is statistically significant (Fig. 6) and that the correlations between model predictions and observations decrease as mean temperatures increase. Given that all our study sites were in the range of highly suitable temperatures for transmission, we would expect this pattern to be even clearer if we included sites with more extreme temperatures.

REVIEWER COMMENTS

Reviewer #1 (Remarks to the Author):

I recognize that the authors have made a considerable effort to respond to my comments. I am therefore wish I could move on and indicate that the manuscript is now ready. Unfortunately, the clearer exposition of what the model can and cannot capture has raised for me a major concern that should be addressed for this work to be convincing (see first comment below). I follow with two other comments that although less fundamental are still important for the paper to be clearer and compelling. These should be my last comments. I really think they cannot be neglected for a paper in Nature Communications.

(1) The model can capture well three main properties of the dynamics. It does so however at a completely different levels of infection than those observed in the real system. Please note that the disagreement here is of orders of magnitude. One could expect that some disagreement would occur between simulated and observed cases, as the model is not calibrated/fitted to the data (as we discussed before). The difference is however stunning in magnitude. This raises the question of whether it is plausible for the model to correct this discrepancy and still maintain "good" prediction of those three properties. That is, there seems to be something fundamentally wrong with the model in terms of infection levels. How do we know that this represents just a relative discrepancy that when fixed would not mess up what the model is actually able to capture. There is no warranty the model is sensible given such high discrepancy. At the least, this should be a serious concern. Would you want to make projections in scenarios with climate change for example that are orders of magnitude off, even if you considered changes to be relative?

(2) The authors have now edited the text to consider more carefully what this modeling approach presents as unique (rather than the mechanistic vs. phenological distinction which was not completely accurate before). Overall the text is now much better in this respect. They still refer in some places to the application of this approach for early-warning and prediction. I would like to re-iterate that there would be no reason to use this kind of approach for seasonal and interannual prediction, rather than a fitted/calibrated model. The latter would perform much better, whether mechanistic or not. The value of this approach would be in scenarios with much longer term projections and in a general model for exploration of questions on interventions . That is, I do not see this approach as appropriate for actual prediction. The model appears to fit timing of the peaks in terms of the month of the peaks but not the interannual variability. That is, the year that exhibits an epidemic would not be well predicted. I agree with the discussion on the complex dynamics of susceptible depletion. Still, I would strongly recommend that the approach is not presented as useful for epidemic prediction.

(3) Finally, the authors started with an interesting contrast between the typical dynamics of epidemics in Kenya and Ecuador. After reading the manuscript, I do not feel I understand why one geographical location is more epidemic than the other. There are results on factors additional to climate. But what have we learned about the fundamental difference between the two continents? How does climate explain the differences (as the title suggest)?

REVIEWER COMMENTS

Reviewer #1 (Remarks to the Author):

I recognize that the authors have made a considerable effort to respond to my comments. I am therefore wish I could move on and indicate that the manuscript is now ready. Unfortunately, the clearer exposition of what the model can and cannot capture has raised for me a major concern that should be addressed for this work to be convincing (see first comment below). I follow with two other comments that although less fundamental are still important for the paper to be clearer and compelling. These should be my last comments. I really think they cannot be neglected for a paper in Nature Communications.

Comment #1: The model can capture well three main properties of the dynamics. It does so however at a completely different levels of infection than those observed in the real system. Please note that the disagreement here is of orders of magnitude. One could expect that some disagreement would occur between simulated and observed cases, as the model is not calibrated/fitted to the data (as we discussed before). The difference is however stunning in magnitude. This raises the question of whether it is plausible for the model to correct this discrepancy and still maintain “good” prediction of those three properties. That is, there seems to be something fundamentally wrong with the model in terms of infection levels. How do we know that this represents just a relative discrepancy that when fixed would not mess up what the model is actually able to capture. There is no warranty the model is sensible given such high discrepancy. At the least, this should be a serious concern. Would you want to make projections in scenarios with climate change for example that are orders of magnitude off, even if you considered changes to be relative?

Response #1: The Reviewer makes an important comment here about the difference in scale between the model predictions and observations. While the Reviewer notes that we would expect some discrepancy between model predictions and observations, we would like to emphasize that we expected substantial differences in the magnitude of predictions and observations because the model produced population level estimates whereas the observations were from a small subsample of the population (e.g., mosquitoes were collected from a random sample of houses and disease cases were recorded for a fraction of symptomatic cases that presented to a hospital/clinic for a disease in which ~80% of cases are asymptomatic). We do understand why the results caused concern for the Reviewer though, so we have adjusted a scaling parameter in the mosquito carrying capacity function to make predictions on a scale similar to the scale of observations. This scaling parameter was already included in the carrying capacity function but was set to predict population-level mosquito abundances. We now note in the manuscript that this scaling parameter should be adjusted to predict for an entire population. After making this adjustment and re-running the models, the order of magnitude is the same for predictions and

observations. Further, the results are very similar to the results we presented previously (updated Figs, 3, 4, 6, and S1-3, Table 2).

Comment #2: The authors have now edited the text to consider more carefully what this modeling approach presents as unique (rather than the mechanistic vs. phenological distinction which was not completely accurate before). Overall the text is now much better in this respect. They still refer in some places to the application of this approach for early-warning and prediction. I would like to re-iterate that there would be no reason to use this kind of approach for seasonal and interannual prediction, rather than a fitted/calibrated model. The latter would perform much better, whether mechanistic or not. The value of this approach would be in scenarios with much longer term projections and in a general model for exploration of questions on interventions. That is, I do not see this approach as appropriate for actual prediction. The model appears to fit timing of the peaks in terms of the month of the peaks but not the interannual variability. That is, the year that exhibits an epidemic would not be well predicted. I agree with the discussion on the complex dynamics of susceptible depletion. Still, I would strongly recommend that the approach is not presented as useful for epidemic prediction.

Response #2: We agree that this type of model is not ideal for prediction when data are available for model fitting; we have edited the text throughout the manuscript for clarification. We emphasize in the Introduction that this type of model might be most useful when trying to understand dynamics outside known conditions (e.g., in a new setting where data are not available for calibration or with future climate conditions that are not represented by data we can collect today; L93-97 and L108-111). Further, we replaced most instances of prediction/predicted/predicts in the discussion with words like “captures”, “generates”, and “reproduces” to better describe the aims of this study of exploring how well the model-generated output matches field observations.

Comment #3: Finally, the authors started with an interesting contrast between the typical dynamics of epidemics in Kenya and Ecuador. After reading the manuscript, I do not feel I understand why one geographical location is more epidemic than the other. There are results on factors additional to climate. But what have we learned about the fundamental difference between the two continents? How does climate explain the differences (as the title suggest)?

Response #3: This is a great point. We have added a paragraph to the Discussion about this topic:

Lines 355-368: “The results of this study shed some light on the influence of climate in driving endemic versus epidemic dengue transmission. Although Ecuador typically experiences seasonal epidemics [6] and Kenya typically experiences low levels of year-round transmission [5], the sites within this study suggest that epidemic transmission is more common in settings with clear

seasonality (e.g., coastal sites) whereas endemic transmission is more common in settings with more climate variability (e.g., inland sites), regardless of country. Coastal sites experienced more regular seasonal climate cycles, likely because oceans buffer climate variability, and this seasonality corresponded with seasonal epidemics. In contrast, the inland sites experienced more day-to-day climate variability, which resulted in more fluctuations in disease cases. As a result, the occurrence and persistence of suitable temperature, rainfall, and humidity conditions enabling outbreaks were less regular in sites with more climate variability. The ability of the model to detect key epidemic characteristics across endemic and epidemic settings indicates that climate plays a major role in driving when outbreaks occur and how long they last.”